



# Benchmarking High-Resolution, Hydrologic Performance of Long-Term Retrospectives in the United States

Erin Towler[1], Sydney S. Foks[2], Aubrey L. Dugger[1], Jesse E. Dickinson[3], Hedeff I. Essaid[4], David Gochis[1], Roland J. Viger[2], and Yongxin Zhang[1]

[1]National Center for Atmospheric Research (NCAR), Boulder, CO, USA
[2]U.S.  Geological Survey (USGS), Lakewood, CO, USA
[3]U.S. Geological Survey, Arizona Water Science Center, Tucson, AZ, USA
[4]U.S. Geological Survey, Moffett Field, CA, USA

*Correspondence to*: Erin Towler (towler@ucar.edu)

**Abstract.** As high-resolution hydrologic models become more widespread, there is a pressing need for systematic evaluation and documentation of their performance. This paper develops and demonstrates a benchmark statistical design that evaluates the long-term performance of two process-oriented, high-resolution, continental-scale hydrologic models that have been developed to assess water availability and risks in the United States (US): the National Water Model v2.1 application of WRF-Hydro (NWMv2.1) and the National Hydrologic Model v1.0 application of the Precipitation-Runoff Modeling System (NHMv1.0). The evaluation is performed on 5,390 streamflow gages from 1983 to 2016 (~33 years) at a daily time step, including both natural and human-impacted catchments, representing one of the most comprehensive evaluations over the conterminous US. The benchmark consists of a suite of metrics for overall performance, their components, and hydrologic-specific signatures. Overall, the model applications show similar performance, with better performance at sites that are less disturbed by human activities, particularly in the West. Both model applications exhibit better performance in the Northeast, Southeast, Pacific Northwest, and high elevation sites in the West. Relatively worse performance is found in the Central region, Southwest, and lower-elevation West. Both models overestimate streamflow volumes at disturbed gages in the West, which could be attributed to not accounting for human activities, such as active management. Both models underestimate flow variability, especially the highest flows. The model applications showed differences in estimation of low flows, with consistent overestimation by the NWMv2.1, and both over- and under-estimation by the NHMv1.0. This benchmark provides a baseline to document performance and measure the evolution of each model application.

## 1 Introduction

Across the hydrologic modeling community, there is a pressing need for more systematic documentation and evaluation of continental-scale land surface and streamflow model performance (Famiglietti et al., 2011). A challenge to hydrologic evaluation stems from the fact that the objectives of hydrologic modeling often vary. Archfield et al. (2015) reviewed how different communities have approached hydrologic modeling in the past, drawing a distinction between hydrologic catchment





modelers whose primary interest has been simulating streamflow at the local to regional scale, versus land surface modelers, who have historically focused on the water cycle as it relates to atmospheric and evaporative processes at the global scale. As modeling approaches have advanced toward coupled hydrologic and atmospheric systems, both perspectives have evolved and
are converging towards the goal of improving hydrologic model performance through more intentional evaluation and benchmarking efforts.

Land surface modeling (LSM) has a rich history of community-developed benchmarking and intercomparison projects (van den Hurk et al., 2011; Best et al., 2015). In addition to comparative evaluations of process-based models, the LSM community
has used statistical benchmarks, which in some cases have been shown to make better use of the forcing input data than state-of-the-art LSMs (Abramowitz et al., 2008; Nearing et al., 2018). The International Land Model Benchmarking (ILAMB) project is an international benchmarking framework developed by the LSM community (Luo et al., 2012) and has been applied to comprehensively evaluate Earth system models, including the categories of biogeochemistry, hydrology, radiation and energy, and climate forcing (Collier et al., 2018). Although hydrology is a component of ILAMB and other LSM benchmarking
efforts, there is a need for closer collaboration with hydrologists to improve hydrologic process representation in these models (Clark et al., 2015).

Hydrologic catchment modeling has begun to move towards large-sample hydrology, an extension of comparative hydrology, where model performance is evaluated for a large sample of catchments, rather than focusing solely on individual watersheds.
This is appealing since evaluating hydrologic models across a wide variety of hydrologic regimes facilitates more robust regional generalizations and comparisons (Gupta et al., 2014). As such, many hydrologic modeling evaluation efforts have begun to encompass larger spatial scales, particularly over the conterminous United States (CONUS). Monthly water balance models have been used to relate CONUS model errors to hydroclimatic variables (Martinez and Gupta, 2010) and for parameter regionalization (Bock et al., 2016). Newman et al. (2015, 2017) and Addor et al. (2018) demonstrate model benchmarking
utilizing a large-sample daily dataset comprised of 600+ small- to medium-sized US basins. Newman et al. (2015) use the coupled Snow-17 snow model and the Sacramento Soil Moisture Accounting Model (SAC-SMA), which is a conceptual hydrologic model with a lumped watershed configuration, to develop the benchmark dataset. In Newman et al. (2017), the Variable Infiltration Capacity (VIC) Model, a more process-oriented hydrologic model that also uses a lumped configuration, is used in an experiment to test increasing model agility against the benchmark dataset created in Newman et al. (2015). Addor
et al. (2018) test predictions from machine learning (random forest) against the conceptual SAC-SMA benchmark dataset from Newman (2015). By using small-to medium-sized basins that are minimally disturbed by human activities, Newman et al. (2015, 2017) and Addor et al. (2018) are able to attribute regional variations in model performance to continental-scale factors. Lane et al. (2019) benchmark the predictive capability of river flow for over 1,000 catchments in Great Britain by using four lumped hydrological models to capture the uncertainty from model structure and parameters. Lane et al. (2019) included
catchments that were both natural and human-impacted catchments, finding poor performance when the water budget is not





closed, such as due to non-modeled human impacts. In terms of high-resolution hydrologic modeling over the CONUS, Tijerina et al. (2021) developed a proof-of-concept for hydrologic model intercomparison, demonstrated by comparing ParFlow-CONUS hydrologic model, version 1.0 and a NOAA U.S. National Water Model configuration of WRF-Hydro, version 1.2. Both models were process-oriented, high-resolution models that incorporate lateral subsurface flow. The evaluation was performed on 2,200 streamflow gages (both impacted by human activities and relatively undisturbed) for a limited domain of CONUS (centered around the Central US) for one-year to investigate model errors.

This study builds on previous large-sample studies by developing a benchmark dataset of two high-resolution, process-oriented models that have been developed to address water issues nationally: the National Water Model v2.1 application of WRF-Hydro (NWM v2.1; Gochis et al., 2020a) and the National Hydrologic Model v1 application of the Precipitation-Runoff Modeling System (NHM v1; Regan et al., 2018). The evaluation is performed on daily streamflow for 5,390 streamflow gages from 1983-2016 (~33 years), including both natural and human-impacted catchments, representing one of the most comprehensive evaluations over the CONUS to date. The benchmark statistical design is comprised of a suite of metrics that include hydrologic-specific metrics, including those measuring overall performance and their components, as well as hydrologic signatures. This paper highlights select results of the benchmarking analysis to document baseline model performance and characterizes overall performance patterns of both models.

## 2 Hydrologic Model Descriptions

### 2.1 The National Water Model v2.1application of WRF-Hydro (NWM v2.1)

The National Center for Atmospheric Research (NCAR) has developed an open-source, spatially distributed, physics-based community hydrologic model, WRF-Hydro (Gochis et al. 2020a; Gochis et al. 2020b), which is the current basis for the National Oceanic and Atmospheric Administration's National Water Model (NWM). The NWM is an operational hydrologic modeling system simulating and forecasting in real-time major water components (e.g., evapotranspiration, snow, soil moisture, groundwater, surface inundation, reservoirs, streamflow) across the CONUS, Hawaii, Puerto Rico, and the U.S. Virgin Islands. We use NWM streamflow simulations from version 2.1 CONUS long-term retrospective analysis (NWMv2.1). The retrospective data are available from public cloud data outlets (such as: https://noaa-nwm-retrospective-2-1-pds.s3.amazonaws.com/index.html). More information on these data is available from the Office of Water Prediction National Water Model page here: https://water.noaa.gov/about/nwm, with additional release notes available here: https://www.weather.gov/media/notification/pdf2/scn20-119nwm_v2_1aad.pdf.

NWMv2.1 is forced by 1-km atmospheric states and fluxes from NOAA's Analysis of Record for Calibration (AORC; National Weather Service, 2021). For the land surface model, NWM v2.1 uses the Noah-MP (Noah-multiparameterization; Niu et al., 2011), which calculates energy and water states and vertical fluxes on a 1-km grid. WRF-Hydro physics-based hydrologic



routing schemes transport surface water and shallow saturated soil water laterally across a 250-m resolution terrain grid and into channels. NWMv2.1 also leverages WRF-Hydro's conceptual baseflow parameterization, which approximates deeper

groundwater storage and release through a simple exponential decay model. The three-parameter Muskingum–Cunge river routing scheme is used to route streamflow on an adapted National Hydrography Dataset Plus (NHDPlus) version 2 (McKay et al., 2012) river network representation (Gochis et al., 2020a). A level-pool scheme is activated on 5,783 lakes and reservoirs across CONUS representing passive storage and releases from waterbodies; however, no active reservoir management is currently included in the NWM. While the operational NWM does include data assimilation, there is no data assimilation

applied in the retrospective simulation. Using the AORC meteorological forcings, NWMv2.1 calibrates a subset of 14 soil, vegetation, and baseflow parameters to streamflow in 1,378 gauged, predominantly natural flow basins. The calibration procedure uses the Dynamically Dimensioned Search algorithm (Tolson and Shoemaker, 2007) to optimize parameters to a weighted Nash-Sutcliffe efficiency (NSE) of hourly streamflow (mean of the standard NSE and log-transformed NSE). Calibration runs separately for each calibration basin, then a hydrologic similarity strategy is used to regionalize parameters

to the remaining basins within the model domain. For the retrospective analysis, NWMv2.1 produces the channel network output (streamflow, velocity), reservoir output (inflow, level, outflow) and groundwater output (inflow, level, outflow) every hour and every 3 hours for land model output (e.g., snow, evapotranspiration, soil moisture) and high-resolution terrain output (shallow water table depth, ponded water depth). For the analysis in this work, hourly streamflow is aggregated to daily averages.

### 2.2 The National Hydrologic Model v1.0 application of the Precipitation-Runoff Modeling System (NHMv1.0)

The U.S. Geological Survey (USGS) has developed the National Hydrologic Model (NHM version 1.0) application of the Precipitation-Runoff Modeling System (PRMS) (Regan et al., 2018). PRMS uses a deterministic, physical-process representation of water flow and storage between the atmosphere and land surface, including snowpack, canopy, soil, surface depression, groundwater storage, and stream networks. Here we use NHM daily discharge simulations from version v1.0

(NHMv1.0) and more specifically, results from the calibration workflow "by headwater calibration using observed streamflow" with the Muskingum-Mann streamflow routing option ("byHRU_musk_obs"; Hay and LaFontaine, 2020).

Climate inputs to the NHMv1.0 are 1-km resolution daily precipitation and daily maximum and minimum temperature from Daymet (version 3; Thornton et al., 2018). The geospatial structure, which defines the default parameters, spatial hydrologic

response units (HRUs) and the stream network, is defined by the geospatial fabric version 1.0 (Viger and Bock, 2014). The NHM is calibrated using a multiple-objective, stepwise approach to identify an optimal parameter set that balances water budgets and streamflow. The first step calibrates for the water balance of each spatial HRU to "baseline" observations of runoff, actual evapotranspiration, soil moisture, recharge, and snow-covered area derived from multiple datasets (Hay and LaFontaine, 2020). The second step considers timing of streamflow by calibration to statistically generated streamflow in

7,265 headwater watersheds having drainage area of less than 3,000 km$^2$. The final step calibrates to observed gaged





streamflow at 1,417 streamgage locations; details of the calibration can be seen in Appendix 1 of LaFontaine et al. (2019). The NHM does not simulate reservoir operations, surface or groundwater withdrawals, or stream releases. The NHM outputs daily streamflow, which is used in the analysis here.

## 3 Benchmark Statistical Design

### 3.1 Metrics

There are many evaluation metrics to choose from to form a benchmark statistical design, and our initial design includes a suite of nine statistical metrics which we refer to as the "standard metric suite" (Table 1). These metrics were chosen through a balance of how to address questions regarding the error between simulated and observed daily streamflow, along with recognition of what the hydrologic community is currently using and familiar with. The standard metric suite includes three

traditional hydrology efficiency metrics, three metrics that characterize interpretable components of overall performance, and three hydrologic signatures. Table 1 includes the statistics and their description, calculation methods, as well as the possible range and perfect value. Metrics were calculated in the statistical software R (R Core Team, 2021), including using the hydroGOF (hydrological goodness of fit) package (Zambrano-Bigiarini, 2020).

The three efficiency metrics included in the standard suite were selected for their precedent, ubiquity, and familiarity in hydrologic evaluation. The purpose of the efficiency metrics is to answer the question of how well the model reproduced the observations in general. The most well-known metric, the Nash-Sutcliffe efficiency, is the normalized mean square error (Nash and Sutcliffe, 1970). The NSE is formulated to emphasize high flows, though it can be artificially high due to seasonality of flows (Schaefli and Gupta, 2007) and models do not necessarily perform well at reproducing high flows when NSE is used for

calibration (Mizukami et al., 2019). To put more emphasis on low flows, we also include the logNSE, where the NSE is computed on log-transformed flows (logNSE; Pushpalatha et al., 2012). The well-known Kling–Gupta efficiency (KGE) is included, also emphasizing high flows, and was developed to address some of the shortcomings of the NSE; it represents a balanced estimate of bias, correlation, and variability (Gupta et al., 2009). Even though both the NSE and KGE are heavily influenced by outliers (Clark et al., 2021), these metrics remain popular and used widely for model calibration and performance

evaluation in the hydrologic community.

Correlation, standard deviation ratio, and percent bias were included because they characterize components of performance that are well-known and are readily understood both within and outside of the hydrologic modeling community. Correlation is calculated using the nonparametric Spearman's rank correlation coefficient (Spearman's r; Helsel et al., 2020). Because daily

streamflow data are highly skewed (violating the normality assumption), Spearman's r is a better estimator of the correlation coefficient than using the linear Pearson estimator (Barber et al., 2019). Correlation quantifies the relationship between modeled and observed, and is often used to assess flow timing, or how well the shape of the hydrograph is reproduced by the



simulations (Tijerina et al., 2021). The ratio of standard deviations between simulations and observed (rSD) shows the relative variability (Gupta et al., 2009; Newman et al., 2017), indicating if the model is over- or under-estimating the variability of the simulated state (in this case, daily streamflow), relative to observations. Percent bias (PBIAS) provides information on if the model is over- or under-estimating the total streamflow volume (based on the entire simulation period).

Three hydrologic signatures defined by Yilmaz et al. (2008), are included to evaluate model performance of different parts of the flow duration curve (FDC). The bias of high flows (the top 2%) is computed to evaluate how well the model captures the watershed response to big precipitation events (PBIAS_HF). To characterize the response to moderate size precipitation events, the bias of the slope of the FDC mid-section, i.e., 20th-70th percentile flows (PBIAS_FDC), is calculated. We note that steeper mid-section FDC slopes are associated with flashier watersheds (i.e., smaller soil storage and more overland flow) and flatter slopes are characterized with slower responding watersheds (Yilmaz et al. 2008). For low flows, the bias of the bottom 30% (PBIAS_LF), offers insight into baseflow performance.

### 3.2 Data

Using the standard metric suite, we evaluate daily simulations from October 1, 1983 to December 31, 2016, or just over 33 years (=~12,100 days). Model simulations are compared to observations at 5,390 USGS stations, referred to as the "cobalt gages" (Foks et al., 2022); stations were included that had a minimum data length of at least 8 years or 2,920 daily observations (i.e., ~25% complete data), though the observations did not need to be continuous (this allows for missing data, including intermittent and/or seasonally operated gages). A subset of the cobalt gages (n = 5,389) also occurs in the Geospatial Attributes of Gages for Evaluating Streamflow, version II dataset (GAGES II; Falcone, 2011), therefore attributes from GAGES-II are used to examine select results. Figure 1 shows the spatial distribution of the gages, along with their designated region; regions are further aggregations of Level II ecoregions as defined by GAGES-II (see Figure 1 caption). Figure 1 shows the uneven distribution of gages: the eastern United States has a dense network of gages, followed by decreasing coverage moving west into the central plains. There is a modest increase in gage density across the intermountain west, and higher coverage along the west coast. Figure 1 also shows the classification, that is, if the site has been characterized as Reference or Non-Reference. Reference gages indicate less-disturbed watersheds, where observations associated with Non-Reference gages have some level of anthropogenic influence (Falcone 2011). Although the Non-Reference gages outnumber the Reference gages – by about 4 to 1 – Reference gages are relatively well-distributed through the regions.

For statistical significance, we conduct pairwise testing, specifically the Wilcoxon signed-rank test. The Wilcoxon signed-rank test is a non-parametric alternative to paired t-test. The Wilcoxon signed-rank test is appropriate here since the metrics (particularly the efficiency metrics) contain outliers and are not necessarily normally distributed.





## 4 Results

Using daily observations and simulations from the NWMv2.1 (Towler et al., 2022a) and NHMv1.0 (Towler et al., 2022b) hydrologic modeling applications, the standard metric suite was calculated for each of the cobalt gages. Here, we provide select results, with a focus on documenting baseline model performance and providing insight towards model diagnostics and 200 development.

Table 2 provides a summary of the results of the standard metric suite for all 5,390 gages, including median values and statistical significance for each statistic and model application. First, we focus on the three efficiency metrics: the medians for the NWMv2.1 are all slightly higher than those of the NHMv1.0, and the differences are statistically significant given the large 205 sample size. The last column includes the correlations for each metric calculated between the model applications. We see that the correlation between the NWMv2.1 and NHMv1.0 are relatively high (>.5), indicating that they are tracking similarly in terms of overall performance. Further, if we examine the correlation between the efficiency metrics by model application, we see that the efficiency metrics are all highly correlated (>0.8; Table 3). This indicates that although users may have preferences for evaluating their model using different efficiencies, these three popular efficiency metrics are providing very similar 210 information in terms of overall performance assessments.

Given this similarity, we document the performance of each model application using a single efficiency, the KGE, as results for NSE and logNSE are similar (corresponding figures and tables for NSE are shown at the end of the Supplemental). KGE has a relatively high correlation of 0.578 between model applications (Table 2), and the NWMv2.1 has a higher median (=0.53) 215 than the NHMv1.0 (=0.46); the slight difference of 0.07 is statistically significant ($p<0.05$) given the large sample size (n=5,390). Figure 2 shows the cumulative density functions for the KGE scores. The NWMv2.1 is performing slightly better for KGE values between 0.0 and 0.8; for instance, for a KGE value of 0.5, 54% of the NWMv2.1 sites have a higher score, while 46% of the NHMv1.0 have a score higher than 0.5. For both models, 8% of sites have a KGE value higher than 0.8. Table 4 bins the KGE scores: for KGE values greater than 0.6, over a third of the total sites are in this category (35% of sites 220 for NHMv1.0 and 41% for NWMv2.1; Table 4). For both models, better performance is achieved in the Northeast, which includes the most sites with KGEs greater than 0.4 (Table 4). Both models also have many sites with poor performance, i.e., where KGE values are less than 0.2 (Figure 2). The sites with KGE values <0.2 contribute to 31% and 27% of the total for the NHMv1.0 and NWMv2.1, respectively. Table 4 shows that most of the sites in this low-fidelity category come from the West, (40% for NHMv1.0 and 47% for NWMv2.1). This can be further investigated by examining the spatial variability of KGE: in 225 the West, more of the poor performing sites are in the arid Southwest and the lower elevation basins in the intermountain West; better performance is seen in the higher elevations in the intermountain West and West Coast, including the Pacific Northwest (Figure 3a for NWMv2.1 and Figure 3b for NHMv1.0). For both models, most of the sites in the 0.2-0.4 range come from the Central region (Table 4), which includes the Central Plains and Western Plains (Figure 1). Figure 3 shows that the Central





region poor performance is concentrated along the plains areas that span from the high plains (i.e., North Dakota) vertically
down through the center of the CONUS (i.e., South Dakota, Nebraska, Kansas, Texas). Performance is more mixed as you
move further east in the Central region (e.g., around the Great Lakes). Relatively good performance is seen in the Southeast.

We also examine model performance by class, that is, if the site has been characterized as Reference or Non-Reference.
Reference gages indicate less-disturbed watersheds, where observations associated with Non-Reference gages have some level
of anthropogenic influence (Falcone 2011). Table 5 shows medians by class: all the medians for the efficiency metrics are
higher for the Reference gages than the Non-Reference gages, noting that there are almost 4 times as many Non-Reference as
Reference gages. For KGE, NWMv2.1 increases from 0.49 to 0.65 and for the NHMv1.0, the increase is from 0.38 to 0.67.
Table 6 shows that the biggest differences between Reference and Non-Reference gages are seen in the West, where for the
NWMv2.1 (NHMv1.0) the median KGE of Reference gages is 0.68 (0.70) and for Non-Reference it is 0.13 (0.14).


Metric differences by site can also be calculated to examine where the model applications are doing relatively better and worse.
Figure 4 shows the spatial distribution of the KGE differences (NWMv2.1 minus NHMv1.0). Positive (purple) colors indicate
the sites where NWMv2.1 has better performance, and negative (orange) colors indicate sites where NHMv1.0 is performing
better. It is noticeable that many of the sites are in the tails, i.e., where KGE differences are +/- 0.25, which occurs because the
efficiency metrics have an unbounded lower range (Table 1). Examining Figures 3 and 4 together shows that for many of the
sites, the biggest differences are occurring at sites that are not performing well to begin with. For example, many of the sites
in the aforementioned Central plains areas show high differences, but this is also an area with poorer performance.

Next we examine the component metrics, starting with Spearman's r. Spearman's r has the highest correlation seen between
models in Table 2 (=0.758), and the NWMv2.1 has a higher median (=0.79) than the NHMv1.0 (=0.75); with a statistically
significant difference given the large sample size. Unlike the efficiency metrics, Spearman's r is a bounded metric (range is
from -1 to 1; Table 1), which can make it easier to examine differences. Taking the difference between model applications,
i.e., NWMv2.1 minus NHMv1, we find that the majority of sites (=3,741) have Spearman's r values within 0.1 of each other
– indicating that the models are performing similarly at most sites. Of greater interest is where the differences in Spearman's
r are greater than +/-0.1; these are shown spatially in Figure 5 and quantified by region in Table 7. The NWMv2.1 has 990
sites where it is doing slightly better, which is defined as a Spearman's r value of between 0.1 and 0.3 higher; 39% of these
are in the Central region, with 21% and 19% in the Northeast and Southeast, respectively. Figure 5 helps visualize where some
of the gains are coming from sub-regionally; for instance, for the Southeast, NWMv2.1 seems to be doing slightly better in
Florida. The NHMv1.0 has 489 sites where it is doing slightly better; 49% of these are coming from the West, and 23% and
21% coming from Central and Southeast, respectively. Similar to what was seen with the efficiency metrics, for Spearman's
r, the Reference sites have higher median values for both model applications (Table 5).



The rSD has one of the lower correlations between models (=0.367 in Table 2), and the NWMv2.1 has a median closer to the perfect score of 1 (=0.910) than the NHMv1.0 (=0.850). Figure 6 breaks the rSD results out by region and model application:
in terms of the medians, both models tend to underestimate the daily flow variability (except for the NWMv2.1 in the West). In the West, both models show a median close to 1, with the NWMv2.1 slightly overestimating and the NHMv1.0 slightly underestimating. For the rest of the regions, the NWMv2.1 has a median closer to 1 for the Central and Southeast, whereas NHMv1.0 has a median closer to 1 in the Northeast. The rSD has a slightly higher value at the Non-Reference gages than at the Reference gages (Table 5); this is because management generally reduces variability.


Next we examine the four percent bias  metrics. Three of the four percent bias metrics are not highly correlated between the NWMv2.1 and NHMv1.0 (Table 2), with PBIAS having the lowest correlation (=0.255). The PBIAS histograms (Figure 7) show that for the NWMv2.1 and NHMv1.0, most of the sites are in the -20 to 20% category (53% and 45%, respectively, Table 8), mainly from the Northeast. Table 8 shows that both models tend to underestimate volumes in the Central region. To
investigate this further, we can examine the PBIAS spatial variability, but only include sites with PBIAS values either greater than 20% or less than -20% (Figure 8). This shows sub-regional differences; for instance, Figure 8 shows that the NHMv1.0 tends to underestimate in the Great Lakes, whereas the NWMv2.0 tends to overestimate in this area. Both models overestimate water volumes in the West. This could be because neither model is capturing active reservoir operations or water extractions (e.g., for irrigation), which is important since water is heavily managed in the West. This is further seen in Table 6, where for
the NWMv2.1 (NHMv1.0) in the Western United States Reference gages have a median PBIAS of 3.1% (0.8%), but Non-Reference gages have a median PBIAS of 44% (20%). The PBIAS_FDC results (Table 9) show that for the CONUS, the +/-20% range bin has the largest number of sites (~40% for both model applications, mainly from the Northeast), followed by underestimation by 20-60% at 30% sites, which is consistent for both model applications. Most of the underestimated sites are in the Central region. Sites where PBIAS_FDC is being over-estimated are generally in the West. Maps of PBIAS_FDC for
biases >+/-20% can be seen in Supplemental Figure 1.

Finally, we can look at results for the high and low flow biases. Results for PBIAS_HF indicate that both models tend to skew towards underestimation of the highest flows (Figure 9), where the percent of CONUS sites with PBIAS_HF < -20% is 60% of the NWMv2.1 sites and 68% of the NHMv1.0 sites (Supplemental Table 1). This is in line with high flow results from
small- to medium-sized catchments examined in Newman et al. (2015) and our previous rSD result that showed that both models tend to underestimate the variability (partially a product of calibrating to NSE, as described in Gupta et al., 2009). Table 6 shows that for PBIAS_HF there is less of a noticeable difference between Reference and Non-Reference sites, and for the NWMv2.1 there is better estimation at the Non-Reference sites, particularly in the West where management is likely reducing variability. The most pronounced difference between the model applications was seen for PBIAS_LF (Figure 10).
The NWMv2.1 tends to overestimate the low flows, as indicated by the positive skew of the histogram. This is broken out by region in Supplemental Table 2: 59% of the NWMv2.1 sites have a PBIAS_LF greater than 20%. On the other hand, the





NHMv1.0 is less skewed, with some over- and under-estimation of the low flows: 37% of the sites have PBIAS_LF >20%, and 22% are <-100%. The NHMv1.0 shows a lower median bias of the low flows, which is statistically significant (Table 2). Looking at the results broken out by class can help to discern if human activities, such as groundwater pumping, are influencing

these results. Looking at the medians broken out by model application and class in Table 6 indicates that model differences may be more important than the Reference versus Non-Reference classification, and that a different attribute (e.g., baseflow index, etc.) could be warranted. Nevertheless, examining the PBIAS_LF results for the reference gages only (Figure 11), we see the NHMv1.0 shows extreme negative flow biases in the Pacific Northwest, California, and Southwest into Texas. The NWMv2.1 shows mostly neutral bias in the Pacific Northwest, similar extreme negative flow bias in Texas, with mixed over-

and under-estimation in other parts of the West. This shows some similarity with Newman et al., (2015), who used a lumped conceptual model to simulate streamflow at small- to medium-sized basins, and found that snowpack-dominated watersheds and central west coast generally had a negative low flow bias. Both the NWMv2.1 and NHMv1.0 are overestimating low flows (positive biases) in most of the Central Plains, as well as the Southeast Coast (especially NWMv2.1). Newman et al. (2015) found that basins in the East, with a smaller seasonal cycle, have a positive low flow bias. In slight contrast, both models have

relatively neutral to negative biases in the Eastern Highlands and Northeast, with more negative biases seen for the NHMv2.1 in the East.

## 5 Discussion and Conclusions

The presented analysis documented baseline model performance and characterized overall performance patterns of two large-sample, long-term hydrologic models for simulating daily streamflow. This analysis is aligned with recent aims of the

hydrologic benchmarking community to put performance metrics in context (Clark et al. 2021); here we provide a lower benchmark to gauge the evolution of the NWMv2.1 and NHMv1.0, two models that have been developed to assess water availability and risks in the United States. The baseline can provide an a priori expectation for what constitutes a "good" model. For instance, as model development activities are undertaken, this can help assess if the overall performance has improved, or if model performance can be tied to a specific application or need, i.e., can we improve the model's representation of low

flows? This is complementary to other model diagnostic and development work that aims to understand model sensitivity and why models improve/degrade with changes. Recent studies have applied sensitivity analyses that consider both parametric and structural uncertainties to identify the water cycle components streamflow predictions are most sensitive to (Mai et al., 2022). Information theory also provides tools that help identify model components contributing to errors (Frame et al. 2021). Further, simple statistical or conceptual models (e.g., Nearing et al., 2018; Newman et al., 2017) could also be used as a benchmark if

applied to the same sites/catchments and time periods.

Overall, the model applications showed similar performance, despite differences in process representations, parameter estimation strategies, meteorological forcings, and space/time discretizations. The efficiency metrics showed that the sites with

 

poor performance tended to be in the Central region, Southwest, and lower elevation intermountain West, and that better
performance was seen in the Northeast, Southeast, higher elevation intermountain West, and Pacific Northwest. The efficiency
and Spearman's r metrics consistently showed that the Reference sites, which are less disturbed by human activities, had better
performance than the Non-Reference sites. It was also notable that despite different forcings (NWMv2.1 is forced by AORC
and NHMv1 is forced by Daymet version 3), the model applications had generally similar performance. Although it was
outside the scope of this study, it would be interesting to explore how forcing biases contribute to streamflow biases.


Results helped to identify potentially missing processes that could improve model performance. PBIAS results showed that
for both models, simulated streamflow volumes are overestimated in the West region, particularly for the sites designated as
Non-Reference. One primary reason for this may be that water withdrawal for human use is endemic throughout the West and
neither model has a thorough representation of these withdrawals. Furthermore, neither model possesses significant
representations for lake and stream channel evaporation which, through the largely semi-arid west, can constitute a significant
amount of water "loss" to the hydrologic system (Friedrich et al., 2018). Lastly, nearly all western rivers are also subject to
some form of impoundment. Even neglecting evaporative, seepage and withdrawal losses from these water bodies, the storage
and timed releases of water from managed reservoirs can significantly alter flow regimes from daily to seasonal timescales
thereby degrading model performance statistics at gaged locations downstream of those reservoirs. Lane et al. (2019) find that
poor model performance occurs when the water budget is not closed, such as when human modifications or groundwater
processes are not accounted for in the models. Model development activities that add management processes can be compared
to the benchmark results here to see if the changes offer improvements.

The model applications showed interesting differences in PBIAS_LF, with the NWMv2.1 overestimating low flows, whereas
the NHMv1.0 both over- and under-estimated them. We note that both models used in the applications benchmarked here have
only rudimentary representation of groundwater processes. Additional attributes (e.g., baseflow or aridity indices) could be
strategically identified to further understand these model errors and differences. Model target applications, which drive model
developer selections for process representation, space and time discretization, and calibration objectives, also have a notable
imprint on the performance benchmarks. The NWMv2.1, with a focus on flood prediction and fast (hourly) timescales, shows
better performance in high-flow-focused metrics, while the NHMv1.0, designed for water availability assessment and slower
(daily) timescales, shows better performance in low-flow-focused metrics.

This study evaluated two state-of-the-art continental-scale models, but the design is general and could be applied to other
hydrologic models, either physically based or statistical. For example, the benchmark statistical design can be used to provide
a regional context for development of refined models in basins of interest (e.g., the U.S. Geological Survey Integrated Water
Science Basins https://www.usgs.gov/mission-areas/water-resources/science/integrated-water-science-iws-basins), where this
design can be used to assess the performance of these basin models relative to national model performance.



Identifying a suite of metrics has an element of subjectivity, but our aim was to identify an initial set of metrics that can be
applied to a wide variety of science questions (e.g., see Table 1.1 in Blöschl et al. 2013) and that build on standard practices
for evaluation of model application performance within the hydrologic community. Different metrics could be more
appropriate for addressing specific scientific questions or modeling objectives, based on the hypothesis-driven development
question being investigated. One limitation of this study is that it does not consider the sensitivity of the NSE and KGE to
sampling uncertainty, which can be large for heavy-tailed streamflow errors (Clark et al., 2021). This could be addressed by
applying bootstrapping methods (Clark et al., 2021). Alternative estimators of NSE and KGE that are more appropriate for
skewed streamflow data (e.g., LBE from Lamontagne et al., 2020) could be added in the future, but currently require separate
treatment of sites with zero streamflow, which was not feasible for this initial statistical benchmark design. As previously
noted, some of the metrics in the benchmark suite include redundant error information; one approach to remedy this has been
put forth by Hodson et al. (2021), where the mean log square error is decomposed to only include independent error
components (see Hodson et al. 2021 for details). This could also be addressed using Empirical Orthogonal Function (EOF)
analysis, which has been done for climate model evaluation (Rupp et al., 2013). Further, this benchmark statistical design is
used to examine pairwise differences, while the Hodson et al. (2021) approach is more conducive to multi-model comparisons.

*Code and data availability*:
NWMv2.1 model data can be accessed through an Amazon S3 bucket, https://registry.opendata.aws/nwm-archive/, and NHM
v1 model data  are available as a USGS data release (Hay and Fontaine 2020).   Results discussed in this publication can be
found in Towler et al. (2022a, 2022b).

*Author Contributions*:
ET and SSF collaborated to develop and demonstrate the benchmark statistical design; ALD, JED, HIE, DG, and RJV
contributed to discussions that shaped the ideas. ET led the results analysis and prepared the original paper. All authors helped
with the editing and revisions of the paper. YZ ran the NWM model and provided the data.

*Competing interests*: The authors declare that they have no conflict of interest.

*Acknowledgements:* We thank the USGS HyTEST Evaluation task team (Robert W. Dudley, Krista A. Dunne, Glenn A.
Hodgkins, Timothy O. Hodson, Sara B. Levin, Thomas M. Over, Colin A Penn, Amy M. Russell, Samuel W. Saxe, Caelan E.
Simeone) for feedback on the development of the benchmark statistical design and the WRF-Hydro group (in particular Ryan
Cabell, Andy Gaydos, Alyssa McCluskey, Arezoo Rafieeinasab, Kevin Sampson, and Tim Schneider. We thank Stacey
Archfield and Andrew Newman for comments on an earlier version of the manuscript. Any use of trade, firm, or product names
is for descriptive purposes only and does not imply endorsement by the U.S. Government.



*Financial Support*: This research is funded by the USGS Integrated Water Prediction Program & NCAR collaboration entitled: A Community Testbed Project for High-Resolution Hydroclimate Science, Simulation, and Application; this includes projects
"HyTest" and "PUMP", as well as input from the IWAAs Program. NCAR is a major facility sponsored by the National Science Foundation (NSF) under Cooperative Agreement 1852977.

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


# Tables

**Table 1. Standard metric suite included in the benchmark statistical design for daily streamflow evaluation. NSE = Nash-Sutcliffe efficiency; KGE = Kling–Gupta efficiency; rSD = ratio of standard deviations between simulations and observed; PBIAS = percent bias; HF = high flows; FDC = flow duration curve; LF = low flows.**

| Category | Statistic | Description | Range (Perfect) | Comments |
|---|---|---|---|---|
| Efficiencies | NSE | Nash-Sutcliffe efficiency (Nash & Sutcliffe, 1970) | -Inf to 1 (1) | Normalized hydrologic metric of overall performance that emphasizes high flows (sensitive to outliers); calculated from NSE in R package hydroGOF. |
| | logNSE | log Nash-Sutcliffe efficiency (Pushpalatha et al., 2012) | -Inf to 1 (1) | Normalized hydrologic metric of overall performance geared toward low flows; calculated from NSE with options FUN=log, epsilon="Pushpalatha2012" in R Package hydroGOF. |
| | KGE | Kling–Gupta efficiency (Gupta et al., 2009) | -Inf to 1 (1) | Normalized hydrologic metric of overall performance geared towards high flows (sensitive to outliers); calculated from KGE in R package hydroGOF. |
| Components | Spearman's r | Spearman's correlation coefficient | -1 to 1 (1) | Nonparametric estimator of correlation for flow shape and timing; calculated from cor function in base R package (method="spearman") |
| | rSD | Ratio of standard deviations | 0 to Inf (1) | Indicates if flow variability is being over- or under-estimated; calculated from rSD in R Package hydroGOF. |
| | PBIAS | Percent bias | -100 to Inf (0) | Indicates if total streamflow volume is being over- or under-estimated; calculated from pbias in R Package hydroGOF. |
| Hydrologic Signatures | PBIAS_HF | Percent bias of flows >=Q98 (Yilmaz et al. 2008) | -100 to Inf (0) | Characterizes response to large precipitation events; calculated using flows >= the 98th percentile flow using pbias in R Package hydroGOF. |
| | PBIAS_FDC | Percent bias of slope of Q20-Q70 FDC (Yilmaz et al. 2008) | -100 to Inf (0) | Characterizes response to moderate precipitation events; calculated from pbiasfdc in R Package hydroGOF. |
| | PBIAS_LF | Percent bias of flows <=Q30 (Yilmaz et al. 2008) | -Inf to 100 (0) | Characterizes baseflow; calculated following equations in Yilmaz et al. (2008) using logged flows <= the 30th percentile (zeros are set to USGS observational threshold of 0.01 cfs). |


**Table 2. Median values of standard metric suite applied to daily streamflows at 5,390 sites in the conterminous United States and correlation (using Spearman's r) between model applications. Bold indicates the median differences are statistically significant as measured by Wilcoxon signed-rank test p-values, grey fill indicates correlation is less than 0.5. NSE = Nash-Sutcliffe efficiency; KGE**
**= Kling–Gupta efficiency; rSD = ratio of standard deviations between simulations and observed; PBIAS = percent bias; HF = high flows; FDC = flow duration curve; LF = low flows.**

| Statistic | Model Median | | Correlation between Models |
|---|---|---|---|
| | NHM v1.0 | NWM v2.1 | |
| NSE | 0.39 | **0.46** | 0.637 |
| logNSE | 0.36 | **0.44** | 0.671 |
| KGE | 0.46 | **0.53** | 0.578 |
| rSpearman | 0.75 | **0.79** | 0.758 |
| rSD | 0.85 | **0.91** | 0.367 |
| PBIAS | -5.1 | **2.1** | 0.255 |
| PBIAS_HF | -32.3 | **-26.7** | 0.370 |
| PBIAS_FDC | **-11.6** | -13.1 | 0.370 |
| PBIAS_LF | **-4.5** | 35.3 | 0.644 |



**Table 3. Correlation (using Spearman's r) between the efficiency metrics for each model application. NSE = Nash-Sutcliffe efficiency; KGE = Kling–Gupta efficiency; NHMv1.0=National Hydrologic Model v1.0; NWMv2.1 = National Water Model v2.1.**

|  | NSE vs KGE | NSE vs logNSE | KGE vs logNSE |
|---|---|---|---|
| NHMv1.0 | 0.89 | 0.80 | 0.79 |
| NWMv2.1 | 0.89 | 0.81 | 0.82 |

**Table 4. For each hydrologic model application, number (percent) of sites in KGE category by region; bold italic indicates maximum category for CONUS; bold indicates maximum number (percent) of sites by KGE category across regions. KGE = Kling–Gupta efficiency; CONUS = conterminous United States; NHMv1.0=National Hydrologic Model v1.0; NWMv2.1 = National Water Model v2.1.**

|  | KGE | CONUS | Region | | | |
|---|---|---|---|---|---|---|
|  |  |  | West | Central | Southeast | Northeast |
| NHMv1.0 | <0.2 | 1668 (31%) | **673 (40%)** | 572 (34%) | 297 (18%) | 126 (8%) |
|  | 0.2-0.4 | 762 (14%) | 171 (22%) | **244 (32%)** | 217 (28%) | 130 (17%) |
|  | 0.4-0.6 | 1050 (19%) | 209 (20%) | 286 (27%) | 261 (25%) | **294 (28%)** |
|  | 0.6-1.0 | *1910 (35%)* | 457 (24%) | 348 (18%) | 437 (23%) | **668 (35%)** |
| NWMv2.1 | <0.2 | 1439 (27%) | **670 (47%)** | 459 (32%) | 249 (17%) | 61 (4%) |
|  | 0.2-0.4 | 582 (11%) | 154 (26%) | **210 (36%)** | 139 (24%) | 79 (14%) |
|  | 0.4-0.6 | 1165 (22%) | 222 (19%) | 314 (27%) | 299 (26%) | **330 (28%)** |
|  | 0.6-1.0 | *2204 (41%)* | 464 (21%) | 467 (21%) | 525 (24%) | **748 (34%)** |

**Table 5. For standard metric suite where the perfect score is 1, median values broken out by Reference (Ref, n= 1,115) and Non-Reference (Non-ref, n= 4,274) gages (one gage was not designated as Ref or Non-ref and is therefore not included); bold indicates higher value for a given model application. NSE = Nash-Sutcliffe efficiency; KGE = Kling–Gupta efficiency; rSD = ratio of standard deviations between simulations and observed; NHMv1.0=National Hydrologic Model v1.0; NWMv2.1 = National Water Model v2.1.**

|  |  | NSE | logNSE | KGE | rSpearman | rSD |
|---|---|---|---|---|---|---|
| NHMv1.0 | Non-ref | 0.34 | 0.22 | 0.38 | 0.73 | **0.86** |
|  | Ref | **0.57** | **0.61** | **0.67** | **0.81** | 0.84 |
| NWMv2.1 | Non-ref | 0.42 | 0.33 | 0.49 | 0.77 | **0.92** |
|  | Ref | **0.56** | **0.65** | **0.65** | **0.84** | 0.87 |





**Table 6. For select metrics, median values broken out by region, as well as Reference and Non-Reference gages (one gage was not designated as Ref or Non-ref and is therefore not included). KGE = Kling–Gupta efficiency; NHMv1.0=National Hydrologic Model v1.0; NWMv2.1 = National Water Model v2.1; PBIAS = percent bias; HF = high flows; LF = low flows.**

| | | | West | Central | Southeast | Northeast |
|---|---|---|---|---|---|---|
| KGE | NHMv1.0 | Non-ref | 0.14 | 0.29 | 0.41 | 0.60 |
| | | Ref | 0.70 | 0.54 | 0.66 | 0.70 |
| | NWMv2.1 | Non-ref | 0.13 | 0.44 | 0.53 | 0.62 |
| | | Ref | 0.68 | 0.46 | 0.67 | 0.70 |
| PBIAS (%) | NHMv1.0 | Non-ref | 20 | -21 | -13 | -2.3 |
| | | Ref | 0.8 | -10 | -5.4 | -3.9 |
| | NWMv2.1 | Non-ref | 44 | 7.5 | 0.5 | -7.7 |
| | | Ref | 3.1 | -3.2 | -6.3 | -8.2 |
| PBIAS_HF (%) | NHMv1.0 | Non-ref | -18 | -42 | -42 | -30 |
| | | Ref | -27 | -39 | -33 | -27 |
| | NWMv2.1 | Non-ref | -1.0 | -28 | -30 | -35 |
| | | Ref | -15 | -44 | -33 | -31 |
| PBIAS_LF (%) | NHMv1.0 | Non-ref | -13 | 20 | -6.3 | -23 |
| | | Ref | -53 | 16 | 14 | -2.7 |
| | NWMv2.1 | Non-ref | 36 | 49 | 51 | 24 |
| | | Ref | -6.5 | 37 | 40 | 10 |


**Table 7. Number of sites by region for which model application performance is categorized as better, based on difference (NWMv2.1 minus NHMv1) in correlation as measured by Spearman's r; sites with differences in the -0.1 to 0.1 category are not included (n=3741); bold italic indicates maximum category for CONUS; bold indicates maximum number (percent) of sites by category across regions. CONUS = conterminous United States; NHMv1.0=National Hydrologic Model v1.0; NWMv2.1 = National Water Model v2.1.**


| Spearman's r Difference | | CONUS | Region | | | |
|---|---|---|---|---|---|---|
| | | | West | Central | Southeast | Northeast |
| NHMv1.0 is better | (-0.7,-0.3] | 73 (1%) | **35 (48%)** | 10 (14%) | 25 (34%) | 3 (4%) |
| | (-0.3,-0.1] | 489 (9%) | **240 (49%)** | 112 (23%) | 102 (21%) | 35 (7%) |
| NWMv2.1 is better | (0.1,0.3] | *990 (18%)* | 203 (21%) | **389 (39%)** | 190 (19%) | 208 (21%) |
| | (0.3,0.7] | 80 (1%) | **36 (45%)** | 30 (38%) | 12 (15%) | 2 (3%) |




**Table 8. For each hydrologic model application, number (percent) of sites in PBIAS category by region; bold italic indicates maximum category for CONUS; bold indicates maximum number (percent) of sites by category across regions. PBIAS = percent bias; CONUS = conterminous United States; NHMv1.0=National Hydrologic Model v1.0; NWMv2.1 = National Water Model v2.1.**

| | PBIAS | CONUS | Region | | | |
| | | | West | Central | Southeast | Northeast |
| --- | --- | --- | --- | --- | --- | --- |
| NHMv1.0 | (-100,-60] | 483 (9%) | 108 (22%) | **245 (51%)** | 104 (22%) | 26 (5%) |
| | (-60,-20] | 1205 (22%) | 196 (16%) | **456 (38%)** | 348 (29%) | 205 (17%) |
| | (-20,20] | *2436 (45%)* | 571 (23%) | 493 (20%) | 578 (24%) | **794 (33%)** |
| | (20,60] | 487 (9%) | **175 (36%)** | 85 (17%) | 94 (19%) | 133 (27%) |
| | (60,100] | 206 (4%) | **107 (52%)** | 37 (18%) | 28 (14%) | 34 (17%) |
| | (100, Inf] | 573 (11%) | **353 (62%)** | 134 (23%) | 60 (10%) | 26 (5%) |
| NWMv2.1 | (-100,-60] | 97 (2%) | 34 (35%) | **39 (40%)** | 17 (18%) | 7 (7%) |
| | (-60,-20] | 627 (12%) | 108 (17%) | **188 (30%)** | 165 (26%) | 166 (26%) |
| | (-20,20] | *2882 (53%)* | 548 (19%) | 708 (25%) | 686 (24%) | **940 (33%)** |
| | (20,60] | 708 (13%) | **279 (39%)** | 197 (28%) | 158 (22%) | 74 (10%) |
| | (60,100] | 321 (6%) | **142 (44%)** | 85 (26%) | 80 (25%) | 14 (4%) |
| | (100, Inf] | 755 (14%) | **399 (53%)** | 233 (31%) | 106 (14%) | 17 (2%) |






**Table 9. For each hydrologic model application, number (percent) of sites in PBIAS_FDC category by region; bold italic indicates maximum category for CONUS; bold indicates maximum number (percent) of sites by category across regions. PBIAS = percent bias; FDC = flow duration curve; CONUS = conterminous United States; NHMv1.0=National Hydrologic Model v1.0; NWMv2.1 = National Water Model v2.1.**


|  | PBIAS_FDC | CONUS | Region | | | |
|---|---|---|---|---|---|---|
|  |  |  | West | Central | Southeast | Northeast |
| NHMv1.0 | (-100,-60] | 376 (7%) | **170 (45%)** | 142 (38%) | 41 (11%) | 23 (6%) |
|  | (-60,-20] | 1594 (30%) | 375 (24%) | **532 (33%)** | 347 (22%) | 340 (21%) |
|  | (-20,20] | *2198 (41%)* | 484 (22%) | 494 (22%) | 528 (24%) | **692 (31%)** |
|  | (20,60] | 597 (11%) | **218 (37%)** | 137 (23%) | 137 (23%) | 105 (18%) |
|  | (60,100] | 236 (4%) | **91 (39%)** | 57 (24%) | 54 (23%) | 34 (14%) |
|  | (100, Inf] | 361 (7%) | **164 (45%)** | 84 (23%) | 92 (25%) | 21 (6%) |
|  | NA | 28 (1%) | 8 (29%) | 4 (14%) | 13 (46%) | 3 (11%) |
| NWMv2.1 | (-100,-60] | 545 (10%) | **225 (41%)** | 207 (38%) | 72 (13%) | 41 (8%) |
|  | (-60,-20] | 1630 (30%) | 341 (21%) | **474 (29%)** | 376 (23%) | 439 (27%) |
|  | (-20,20] | *2158 (40%)* | 553 (26%) | 476 (22%) | 487 (23%) | **642 (30%)** |
|  | (20,60] | 535 (10%) | **168 (31%)** | 142 (27%) | 164 (31%) | 61 (11%) |
|  | (60,100] | 229 (4%) | **91 (40%)** | 62 (27%) | 54 (24%) | 22 (10%) |
|  | (100, Inf] | 241 (4%) | **107 (44%)** | 69 (29%) | 53 (22%) | 12 (5%) |
|  | NA | 52 (1%) | 25 (48%) | 20 (38%) | 6 (12%) | 1 (2%) |





# Figures

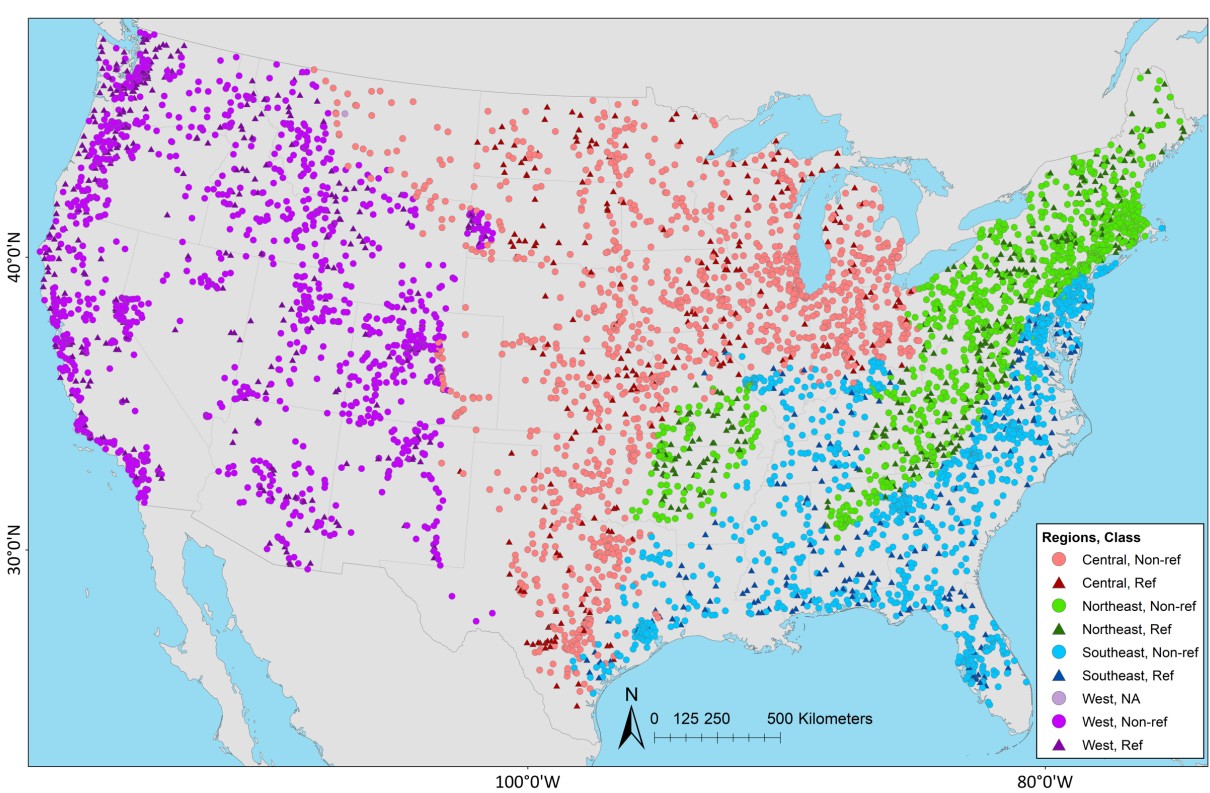

**Figure 1: Site locations used in evaluation (n=5,390), including regions and classification. Regions were further combinations of aggerated ecoregions defined by Falcone (2010): Central (n=1,450) includes Central Plains, Western Plains, and Mixed Wood Shield; Northeast (n=1,218) includes Northeast and Eastern Highlands; Southeast (n=1,212) includes South East Plains and South East Coastal Plains; and West (n=1,510) includes Western Mountains and West Xeric. Classifications are from Falcone (2010): Reference (Ref, n= 1,115) and Non-Reference (Non-ref, n= 4,274); one gage was not designated (NA, n=1).**





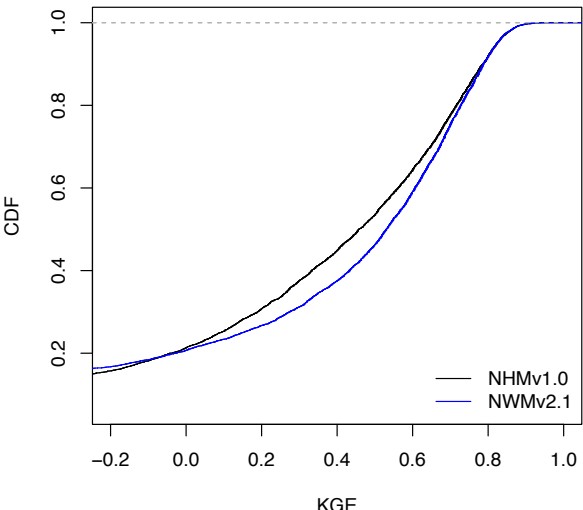

**Figure 2: Cumulative density functions (CDFs) for model Kling-Gupta efficiency (KGE) values based on daily streamflow at U.S.**
**Geological Survey (USGS) gages for National Water Model v2.1 (NWMv2.1) and National Hydrologic Model v1.0 (NHMv1.0).**

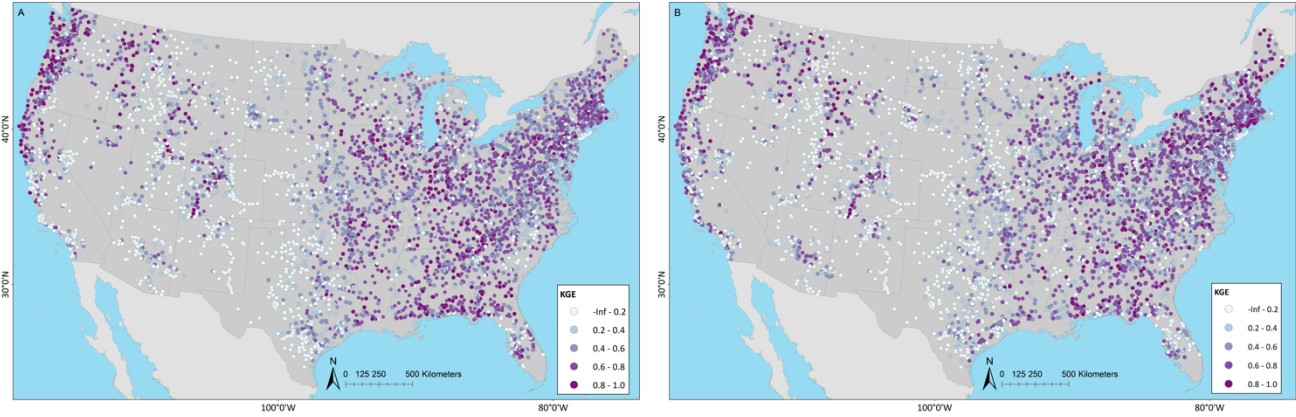

**Figure 3: Kling–Gupta efficiency (KGE) based on daily streamflow at U.S. Geological Survey (USGS) gages for (A) National Water Model v2.1 (NWMV2.1) and (B) National Hydrologic Model v1.0 (NHMv1.0).**






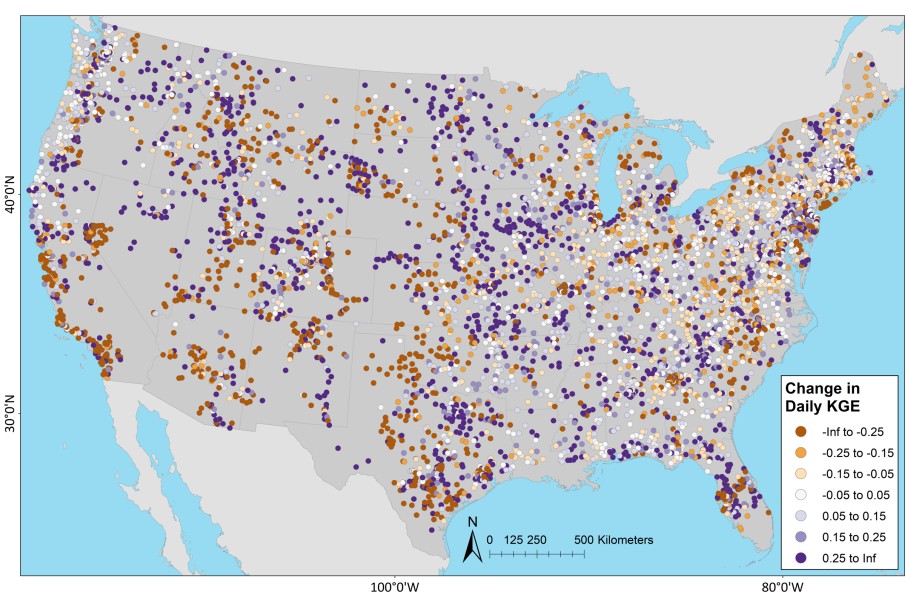

**Figure 4: Difference in Kling–Gupta efficiency (KGE), National Water Model v2.1 (NWMv2.1) minus National Hydrologic Model v1.0 (NHMv1.0), based on daily streamflow at U.S. Geological Survey (USGS) gages; negative (orange) indicates where NHMv1.0 has a higher (better) KGE, positive (purple) indicates NWMv2.1 has a higher (better) KGE.**

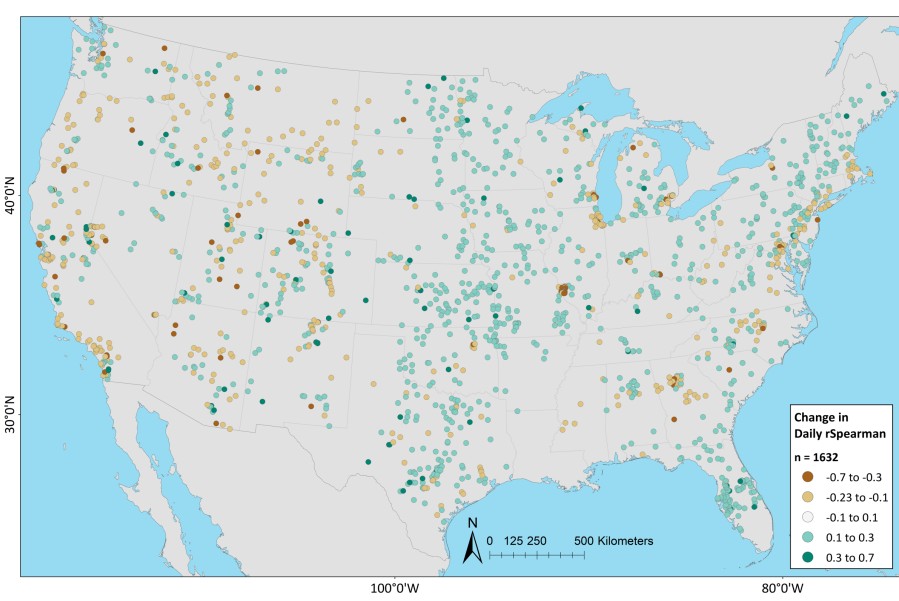


**Figure 5: Difference in Spearman's r, National Water Model v2.1 (NWMv2.1) minus National Hydrologic Model v1.0 (NHMv1.0); negative (brown) indicates where NHMv1.0 is doing better, positive (green) indicates where NWMv2.1 is doing better. Only sites with values >0.1 and <-0.1 are plotted (n=1,632).**





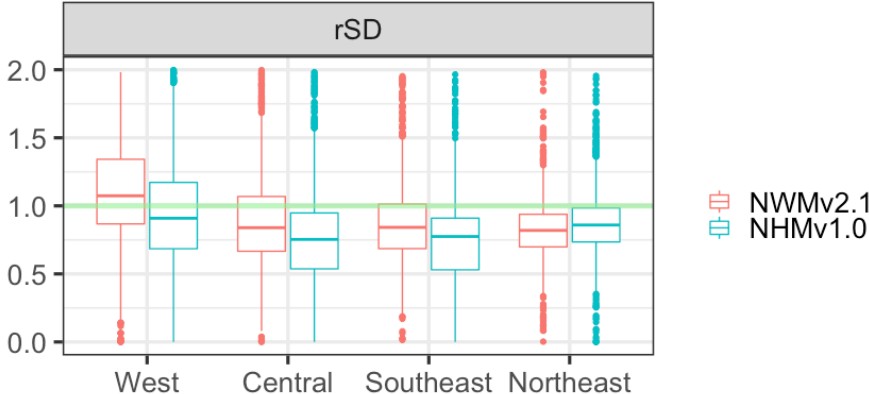

**Figure 6: Standard deviation ratio (rSD) based on National Water Model v2.1 (NWMv2.1) and National Hydrologic Model v1.0 (NHMv1.0) daily streamflow at U.S. Geological Survey (USGS) gages grouped by region. Results are shown as box plots, where the box represents the 25th and 75th percentile, the horizontal line is the median, and the upper and lower whiskers represent up to 1.5 times the interquartile range (IQR), respectively. Points outside the box and whiskers are considered outliers based on the 1.5 times IQR threshold.**


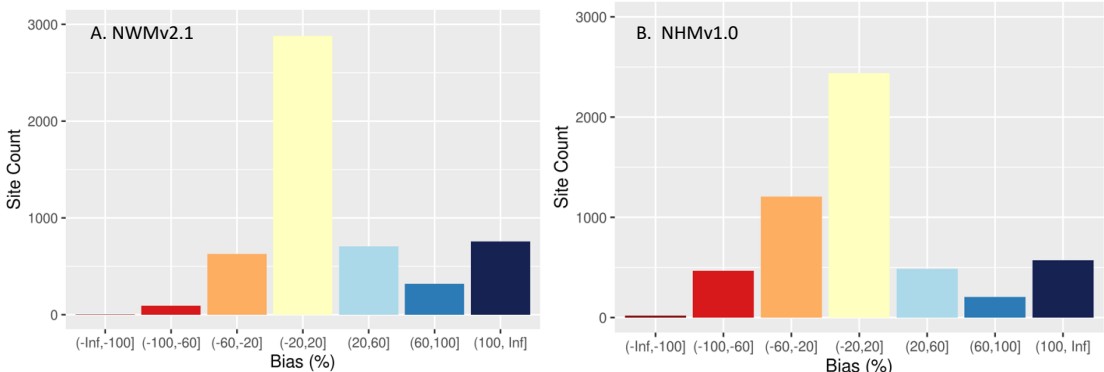

**Figure 7: Percent bias (PBIAS) histograms for (left, A) National Water Model v2.1 (NWMv2.1) and (right, B) National Hydrologic Model v1.0 (NHMv1.0) daily streamflow at U.S. Geological Survey gages in the conterminous United States. Inf = infinity.**




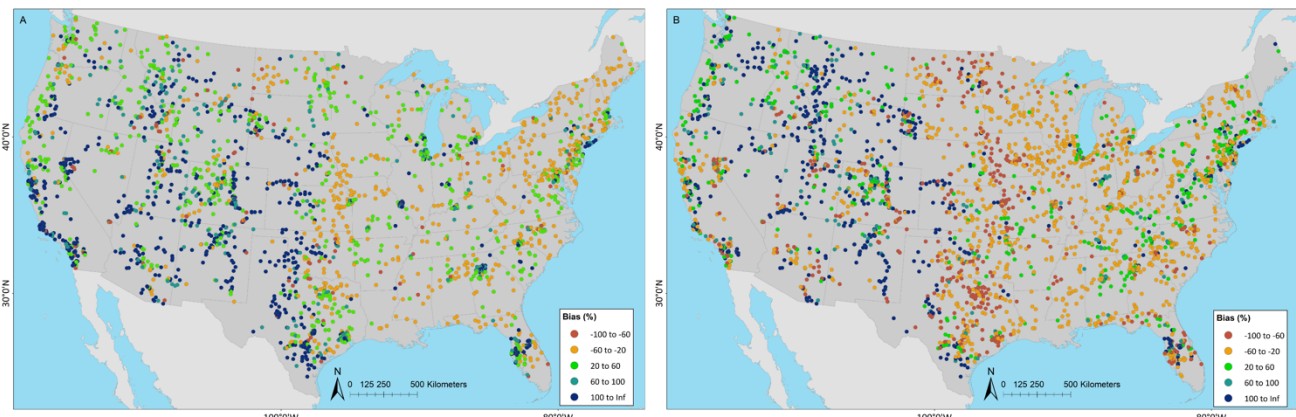

**Figure 8: Percent bias (PBIAS) maps for National Water Model v2.1 (NWMv2.1) (left; A) and National Hydrologic Model v1.0 (NHMv1.0) (right; B), where PBIAS >20% or <-20%. Cooler colors are where model application is overestimating volume and warmer colors are where model is underestimating volume.**

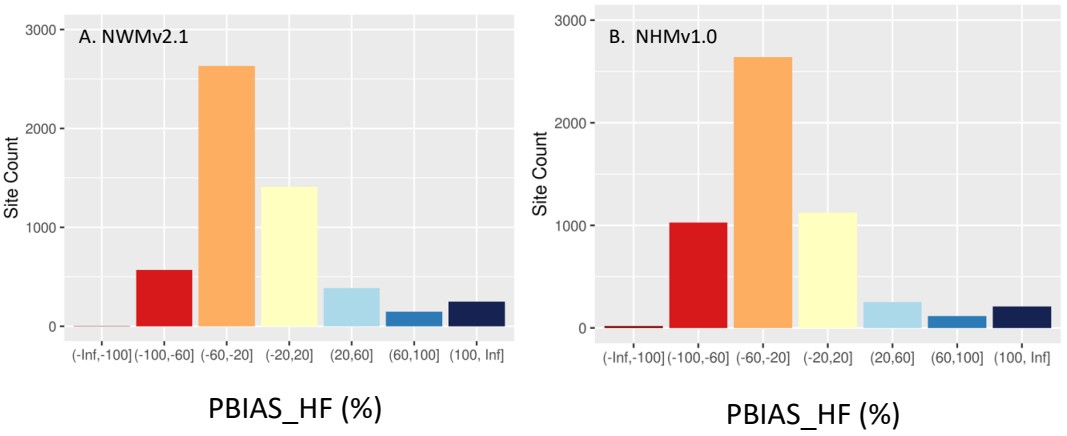

**Figure 9: Percent bias in high (>2%) flows (PBIAS_HF) for (left, A) National Water Model v2.1 (NWMv2.1) and (right, B) National Hydrologic Model v1.0 (NHMv1.0) daily streamflow at U.S. Geological Survey gages in the conterminous United States. Inf = infinity.**





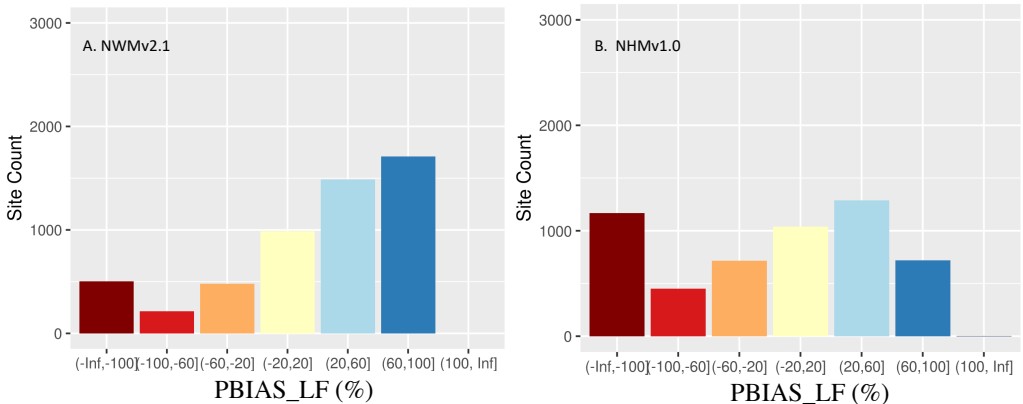

**Figure 10: Percent bias in low (<30%) flows (PBIAS_LF) for (left, A) National Water Model v2.1 (NWMv2.1) and (right, B) National Hydrologic Model v1.0 (NHMv1.0) daily streamflow at U.S. Geological Survey gages in the conterminous United States. Inf = infinity.**


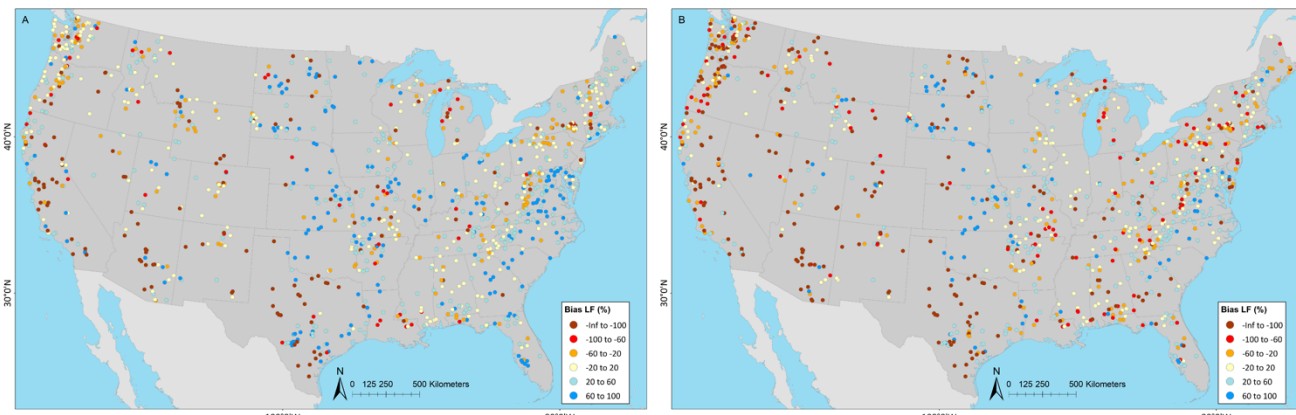

**Figure 11: For reference sites only, percent bias in low (<30%) flows (PBIAS_LF) for National Water Model v2.1 (NWMv2.1) (left, A) and National Hydrologic Model v1.0 (NHMv1.0) (right, B). Cooler colors are where model is overestimating volume and warmer colors are where model is underestimating volume.**
