# Peer review of "Benchmarking High-Resolution, Hydrologic Model Performance of Long-Term Retrospectives in the Contiguous United States"

_Hydrology and Earth System Sciences, 2022_

## Author Comment (AC2)

**General Response for All Reviewers**:

We thank all three Referees for their reviews, which have been addressed to improve the paper. Though we provide a point-by-point response to each reviewer, we also have put together this General Response, articulating the main changes that were made to the manuscript, which should be read first. We focus on main changes in four of the sections, including: (i) Introduction, (ii) Evaluation Approach, (iii) Discussion, and (iv) Results. We refer to this in our point-by-point responses as needed. Further, although we have not finalized our manuscript revision, we have added relevant drafted excerpts here, followed by the updated Tables and Figures, to give the Referees a better sense how these changes were integrated.

(i) **Introduction**: *Added citations and revised to better articulate our contribution*. We thank the reviewers for their suggestions from the literature to provide more context for our work. In addition, we have edited the final paragraph to better highlight our contribution. We provide the relevant revised draft excerpts from the last 3 paragraphs of the Introduction here:

[revised manuscript text omitted]

(ii)     **Evaluation Approach:** *Reduced number of evaluation metrics and focus on KGE, added climatological benchmarks, used climatological benchmark as threshold to screen results for more targeted analysis.*
Based on the Referee suggestions, we focus on KGE, removing the other efficiency metrics (NSE and logNSE), and include the components of KGE (we replaced Spearman's rank with linear r), and focused on only two of the hydrologic signatures (we removed PBIAS_FDC). See Table 1 for updated metrics. We also provide performance context by using climatological benchmarks outlined in Knoben et al. (2019) and Knoben et al. (2020). As suggested by the Referees, we also provide context for the interpretation of the KGE scores; we now note that a lower benchmark must be specified (Pappenberger et al., 2015; Schaefli and Gupta, 2007; Seibert, 2001; Seibert et al., 2018). Further, as pointed out by the Referees, the KGE does not include a built-in lower benchmark in its formulation, but Knoben et al. (2019) show that models with KGE scores higher than −0.41 contribute more information than the mean flow benchmark. We also point to Knoben et al. (2020), who show that it is more robust to define a lower benchmark that considers seasonality. Hence, a reference time series based on the average and median flows for each day-of-year is used to calculate a lower KGE value which serves as a climatological (lower) benchmark; these are referred to as AvgDOY and MedDOY, respectively. We used this threshold to further target out analysis (see Results, revised below).

(iii)    **Discussion**: *Revised discussion to address several points raised by Referees.*
The Referees raised several useful points, many of which we address in the Discussion now; these are pointed out in the point-by-point responses. But in general, we tightened up our discussion of using the benchmark to determine what a "good" model is, and discussed some of the updated results and what they might mean for model diagnostics and development.

(iv)     **Results:** *Replaced tables with cumulative density functions, anchor KGE results based on climatological benchmarks, use climatological KGE benchmark to focus on underperforming sites.*
As indicated above, the Referees offered several suggestions that helped to re-shape the manuscript and its results. We provide the updated Results section here, followed by the draft Figures and Tables to show the Referees how these changes manifested in the updated manuscript.

*4 Results*

*Using daily observations and model simulations, the evaluation metrics from Table 1 are calculated for each of the gages for the NWMv2.1 (Towler et al., 2022a) and NHMv1.0 (Towler et al., 2022b) hydrologic modelling applications. KGE is also calculated using daily observations and day-of-year averages (AvgDOY) and medians day-of-year (MedDOY) to produce a seasonal KGE benchmark for each site.*

*KGE scores for the benchmarks and models can be seen as a cumulative density functions (CDFs; Figure 2), and Table 2 quantifies the percent of sites less than or greater than select KGE scores. First, the seasonal benchmarks and model KGE scores can be compared to the mean flow benchmark (i.e., KGE <-0.41; Knoben et al. 2019): for the MedDOY benchmark, 18% of sites have lower scores, and using the AvgDOY benchmark is always better than using the mean flow. 
[revised manuscript text omitted]

[Figure]

**Supplemental Figure 1. For the National Water Model v2.1 (NWMv2.1) and the National Hydrologic Model v1.0 (NHMv1.0), the number of models where the KGE value is greater than the AvgDOY; both models are better (n=3396), one model is better (n = 1083), or neither model is better (n=911).**

[Figure]

**Supplemental Figure 2. Difference between the Kling–Gupta efficiency (KGE) from the National Water Model v2.1 (NWMv2.1) and the seasonal benchmark based on the average day-of-year flows (AvgDOY); negative (orange) indicates where AvgDOY has a higher (better) KGE, positive (purple) indicates that the NWMv2.1 has a higher (better) KGE.**

[Figure]

**Supplemental Figure 3. Difference between the Kling–Gupta efficiency (KGE) from the National Hydrologic Model v1.0 (NHMv1.0) and the seasonal benchmark based on the average day-of-year flows (AvgDOY); negative (orange) indicates where AvgDOY has a higher (better) KGE, positive (purple) indicates that the NHMv1.0 has a higher (better) KGE.**

[Figure]

**Supplemental Figure 4: Normalized histograms of PBIAS for National Water Model v2.1 (NWMv2.1, top) and National Hydrologic Model v1.0 (NHMv1.0, bottom), for all sites (left) and for sites where the model's KGE score is less than the average day-of-year flow benchmark (right).**

[Figure]

**Supplemental Figure 5: Percent bias of high flow (PBIAS_HF; i.e., exceeding top 2%) maps for National Water Model v2.1 (NWMv2.1) (A) and National Hydrologic Model v1.0 (NHMv1.0) (B), for sites where the KGE score is less than the average day-of-year flow (AvgDOY) benchmark. Cooler colors are where model application is overestimating high flow bias and warmer colors are where model is underestimating high flow bias.**

[Figure]

**Supplemental Figure 6: Normalized histograms of Percent bias of high flow (PBIAS_HF; i.e., exceeding top 2%) for National Water Model v2.1 (NWMv2.1, top) and National Hydrologic Model v1.0 (NHMv1.0, bottom), for all sites (left) and for sites where the model's KGE score is less than the average day-of-year flow benchmark (right).**

[Figure]

**Supplemental Figure 7: Normalized histograms of percent bias of low flow (PBIAS_LF, flows below 30% percentile) for National Water Model v2.1 (NWMv2.1, top) and National Hydrologic Model v1.0 (NHMv1.0, bottom), for all sites (left) and for sites where the model's KGE score is less than the average day-of-year flow benchmark (right).**

[Figure]

**Supplemental Figure 8: ratio of standard deviation (rSD) maps for National Water Model v2.1 (NWMv2.1) (A) and National Hydrologic Model v1.0 (NHMv1.0) (B), for sites where the KGE score is less than the average day-of-year flow (AvgDOY) benchmark. Cooler colors are where model application is overestimating variability and warmer colors are where model is underestimating variability.**

[Figure]

**Supplemental Figure 9: Normalized histograms of standard deviation ratio (rSD) for National Water Model v2.1 (NWMv2.1, top) and National Hydrologic Model v1.0 (NHMv1.0, bottom), for all sites (left) and for sites where the model's KGE score is less than the average day-of-year flow benchmark (right).**

[Figure]

**Supplemental Figure 10: Pearson's correlation coefficient (r) for National Water Model v2.1 (NWMv2.1) (A) and National Hydrologic Model v1.0 (NHMv1.0) (B), for sites where the KGE score is less than the average day-of-year flow (AvgDOY) benchmark.**

**Equations:**

The percent bias in the high flows (PBIAS_HF) is defined as (Yilmaz et al. 2008):

$$PBIAS_{HF} = \frac{\sum_{h=1}^{H}(S_h - O_h)}{\sum_{h=1}^{h} O_h}$$

Where h = 1, 2,… H are the low flow indices for flows with exceedance probabilities lower than 0.02.

The percent bias in the low-flow (PBIAS_LF) is defined as (Yilmaz et al. 2008):

$$PBIAS_{LF} = -1 \cdot \frac{\sum_{l=1}^{L}[log(S_l) - log(S_L)] - \sum_{l=1}^{L}[log\,(O_l - log(O_L)]}{\sum_{l=1}^{L}[log(O_l) - log(O_L)]} \times 100$$

where l = 1, 2,…L is the flow value index in the low-flow segment (0.7–1.0 flow exceedance probabilities) of the flow duration curve and L is the minimum flow index.

---

## Author Comment (AC3)

**Referee #1.**

Review of "Benchmarking High-Resolution, Hydrologic Performance of Long-Term Retrospectives in the United States" by Towler et al.

**Summary**

In this paper, performance of the National Water Model (NWM) v2.1 and the National Hydrologc Model (NHM) v1.0 is evaluated over the United States. These models are different in their internal structure, use different calibration approaches and are run with different meteorological inputs, but are similar in the sense that both are run over a high-resolution spatial grid. Model performance is evaluated with the help of 9 different metrics (e.g. Nash-Sutcliffe, PBIAS) that are calculated using observations and model simulations at 5390 streamflow gauges. Attention focuses most on median values in the 5390-member sample, and on differences between both models in various broad regions across the US. There are some recommendations on how to improve both models; most notably by updating the model structures to account for human water use and the impact of lakes and reservoirs.

**General comments**

Having read this paper, I must admit that I am not entirely sure whether HESS is the right venue for this. Various sentences suggest that this publication is intended as a benchmark for further development of the NWM and HWM. For example:

- [line 25] "This benchmark provides a baseline to document performance and measure the evolution of each model application"

- [line 80] "This paper highlights select results of the benchmarking analysis to document baseline model performance and characterizes overall performance patterns of both models."

- [line 198] "Here, we provide select results, with a focus on documenting baseline model performance and providing insight towards model diagnostics and development."

- [line 315] "here we provide a lower benchmark to gauge the evolution of the NWMv2.1 and NHMv1.0".

This is a great goal that I think should be the standard in any model development exercise (as it is in many other fields), but this kind of benchmarking is of limited interest to anyone who does not actively work with these models. A technical report instead of a journal publication might be more appropriate.

We disagree in that this shows how a benchmarking approach can be used for additional modeling applications.

To appeal to a wider (international) journal audience, the proposed benchmarking approach should be of general interest and I think in its current shape it fails to be that.

**General Reponse to Referee #1**: We thank the Referee for these comments. We note that we had an initial "Short Response" to your general comments, and now we have fleshed out our response in the "General Response for All Referees", which should be read first. We also respond here to each individual comment in this point-by-point response. As noted in the General Response, we have made major revisions to the (i) Introduction, (ii) Evaluation Approach (iii) Discussion and (iv) Results; this has made our study now of more general interest, and suited to a HESS Research Article, and we appreciate the Referee comments that helped us to achieve this end.

My main concerns are that:

1. The selected benchmarking metrics are too one-sided: out of the 9 metrics, 7 either include or are some form of model bias metric. Multiple other relevant aspects of hydrographs and model performance are not captured by these metrics.

Please see "Evaluation Approach" in the General Response. As indicated, we now focus on KGE and reduce the number of metrics examined (see Table 1 in General Response, and included here):

Table 1. Standard metric suite included in the daily streamflow evaluation. KGE = Kling–Gupta efficiency; Pearson's r = linear correlation; rSD = ratio of standard deviations between simulations and observed; PBIAS = percent bias; HF = high flows; LF = low flows.

| Statistic | Description | Range (Perfect) | Comments |
|---|---|---|---|
| KGE | Kling–Gupta efficiency (Gupta et al., 2009) | -Inf to 1 (1) | Normalized hydrologic metric of overall performance geared towards high flows (sensitive to outliers); calculated from KGE in R package hydroGOF. |
| Pearson's r | Pearson's correlation coefficient | -1 to 1 (1) | Pearson (linear estimator) of correlation; calculated from rPearson in R Package hydroGOF. |
| rSD | Ratio of standard deviations | 0 to Inf (1) | Indicates if flow variability is being over- or under-estimated; calculated from rSD in R Package hydroGOF. |
| PBIAS | Percent bias | -100 to Inf (0) | Indicates if total streamflow volume is being over- or under-estimated; calculated from pbias in R Package hydroGOF. |
| PBIAS_HF | Percent bias of flows >=Q98 (Yilmaz et al. 2008) | -100 to Inf (0) | Characterizes response to large precipitation events; calculated using flows >= the 98th percentile flow using pbias in R Package hydroGOF. |
| PBIAS_LF | Percent bias of flows <=Q30 (Yilmaz et al. 2008) | -Inf to 100 (0) | Characterizes baseflow; calculated following equations in Yilmaz et al. (2008) using logged flows <= the 30th percentile (zeros are set to USGS observational threshold of 0.01 cfs). |

In the Discussion, we now also acknowledge that we focus on magnitude, since one of the main purposes of the model evaluation was to assess the suitability of the models for water availability studies. However, we now note that magnitude is only one aspect of streamflow, and that different metrics for other categories (e.g., frequency, duration, rate of change, etc), could be more appropriate for addressing specific modeling objectives.

2. There is no clear way to relate a model's performance on this set of metrics to concrete suggestions for improvement of the model, because it is practically impossible to trace the scores a model obtains on these metrics to how well the model simulates a given hydrological process (though I appreciate that this is not an easy thing to do).

We agree that this is a difficult endeavor. Nevertheless, we liked the suggestion of adopting the climatological benchmark of Knoben et al. 2020, since it offers a concrete goal for model development. Further, by screening our results to look mainly at the underperforming sites (I.e., sites that have KGE values below the climatological benchmark), we were able to come up with several hypothesis as to why, which could be useful for model development.

3. The model results are presented in a vacuum: there is only very limited discussion of existing literature on benchmarking, there is no comparison of the performance of these two models to the performance of earlier modeling efforts across this domain, and there is no discussion about how high a model must score on any of the 9 metrics to be considered a good/plausible/acceptable/etc model.

As seen in our General Response, we have done a major revision, and adopted the suggested approach of comparing with a climatological seasonal benchmark, and using this as a threshold to screen the results.

4. There is almost no guidance (or better yet, software) available for a reader who might want to apply this benchmarking approach to their own simulations, beyond a table that shows references for the 9 metrics and a CSV file that contains the list of gauge IDs.

As mentioned in the initial Short Response, most of the metrics are straightforward to calculate (we use HydroGOF in R), and we have also added the Equations to the manuscript. Based on other comments below, we clarify the utility of the published metrics, and we update them by adding the climatological mean/median benchmark for each site.

I believe that these issues can be addressed to a certain extent (see specific comments below), but in its current shape this manuscript mostly describes what performance scores two arbitrary models obtain on a limited selection of model performance metrics, without any context for those scores whatsoever. I don't think that's enough to warrant publication in HESS.

Please see General Response for the Major Revisions applied to address the Reviewer's comments.

**Specific comments**

l12. "a benchmark statistical design" - It's unclear to me what this means.

We have removed this term from the text.

l90. "https://noaa-nwm-retrospective-2-1-pds.s3.amazonaws.com/index.html" - The NWM docs (https://water.noaa.gov/about/output_file_contents) seem to say that output files are in netCDF4 format, but if I follow this link all I can find is .comp files. What are these files and how can a reader open/use them?

The output files are in netCDF files, they are just tagged as ".comp" because they are compressed. We will add to the manuscript: "(e.g., compressed netcdf files can be found at:…". The netCDF package in R allows for opening and viewing of netcdf files, but a reader can use a variety of programs to open these files.

l105. "Using the AORC meteorological forcings, NWMv2.1 calibrates a subset of 14 soil, vegetation, and baseflow parameters to streamflow in 1,378 gauged, predominantly natural flow basins. The calibration procedure uses the Dynamically Dimensioned Search algorithm (Tolson and Shoemaker, 2007) to optimize parameters to a weighted Nash-Sutcliffe efficiency (NSE) of hourly streamflow (mean of the standard NSE and log-transformed NSE). Calibration runs separately for each calibration basin, then a hydrologic similarity strategy is used to regionalize parameters to the remaining basins within the model domain." - This needs a reference to indicate where a reader can find further details about this procedure.

At this time, there is no publication to reference on this, but authors on this publication provided additional details. Based on this and other Referee comments, we have added the calibration periods to the model descriptions. For the NWMv2.1, the calibration period was from water years 2008 – 2013, and 2014-2016 was used for validation. For the NHMv1.0, the calibration period included the odd water years from 1981-2010, and the even water years from 1982-2010 were used for validation.

l113. "For the analysis in this work, hourly streamflow is aggregated to daily averages." - Looking at a snapshot of the USGS gauges used for this evaluation approach, observations seem to be available at a sub-daily resolution. Given that the model is run at a 3-hr resolution, and it is known that hydrologic processes of interest can show strong diurnal variation (e.g. evaporation, snowmelt), why are observations and simulations aggregated to daily values?

Not all of the gages contain sub-daily records for the temporal extent of interest (1983-2016). Additionally, the NHM only can simulate streamflow at the daily timestep and comparison of these two models on different timesteps was not appropriate. For the benefit of the Referee, we note that other internal evaluations of the NWM have been conducted hourly, but that wasn't the focus for this study.

l148. "The NSE is formulated to emphasize high flows" - This statement seems to contradict the last part of this sentence: "models do not necessarily perform well at reproducing high flows when NSE is used for calibration". Suggest to rephrase this.

In our revision, we have removed NSE (and logNSE) to focus on KGE results, and the new climatological benchmark comparison. As such, this sentence has been removed.

l156. "Correlation, standard deviation ratio, and percent bias" - These three are (almost) the constitutive components of the KGE metric, and also appear in the NSE (see e.g. the decomposition of RMSE by Murphy, 1988, https://doi.org/10.1175/1520-0493(1988)116%3C2417:SSBOTM%3E2.0.CO;2). There is

likely value in looking at these individual components compared to the aggregated efficiency scores, but this section should state that these metrics are not independent from NSE and KGE.

Thank you for raising this point. In our revision, we are focusing on KGE, and agree that there is value in looking at its constituents. We now look at the components of linear correlation (as opposed to in the previous draft, we looked at Spearman's correlation), standard deviation ratio, and percent bias. These equations are now included in the manuscript.

l167. "Three hydrologic signatures defined by Yilmaz et al. (2008)" - There are many possible signatures one could chose from and these are sometimes divided into five separate categories (magnitude, frequency, duration, timing and rate of change; e.g. Olden & Poff, 2003, dx.doi.org/10.1002/rra.770). More recently, McMillan (2022; dx.doi.org/10.1002/hyp.14537) created a signature taxonomy that relates signatures to specific hydrologic processes. The selected signatures here exclusively address the magnitude component, without explaining why these other components are not addressed or how a model's performance on any of these signatures might inform which of the model's process representations needs to be improved.

More generally, out of the 9 presented metrics, 7 metrics are either some form of bias or include a bias component. This seems insufficient spread to me for a "standard metric suite". I believe this selection needs to be expanded quite a bit before these metrics can start to be used for comprehensive model benchmarking.

We appreciate this comment and these references, this was part of the impetus towards our Major Revision (see General Response). We have added draft material to the Discussion to address this point explicitly in the paper: "Identifying a suite of evaluation metrics has an element of subjectivity, but our aim was to focus on streamflow magnitude, since the purpose of the model evaluation effort was for water availability applications. However, magnitude is only one aspect of streamflow, and different metrics for other categories (e.g., frequency, duration, rate of change, etc) could be more appropriate for addressing specific scientific questions or modeling objectives. Recently, McMillan (2019) links hydrologic signatures to specific processes using only streamflow and precipitation. Interestingly, McMillan (2019) does not find many signatures that relate to human alteration; however, in this paper, streamflow bias metrics are found to be useful in this regard."

l170. "big precipitation" - This might be inaccurate phrasing in the case of colder catchments, where flow events might originate from snow/ice melt and not directly from individual precipitation events.

Thank you for this comment. We have added "big precipitation or melt events".

l178. "Foks et al., 2022" - The .csv file in this reference misses leading zeroes for station numbers, which makes searching for them somewhat difficult on the USGS website (https://waterdata.usgs.gov/nwis/uv?referred_module=sw&search_criteria=search_site_no&search_criteria=site_tp_cd&submitted_form=introduction). E.g. searching for station 1011000 yields no results with the default "exact match" option, whereas 01011000 does show a result. If possible, updating this resource could help others. Adding some guidance on how to obtain these observations in a reasonably efficient manner would be good too.

We are not sure what software you used to open the CSV files, but a text-editor such as Rstudio, Notepad++, Visual Studio are common to use for opening CSV file formats so that leading zeros are observable. Microsoft Excel truncates data types it assumes are numeric values. The metadata accompanying this release has information regarding the leading zero for the station IDs.

l191. "For statistical significance, we conduct pairwise testing, specifically the Wilcoxon signed-rank test. The Wilcoxon signed-rank test is a non-parametric alternative to paired t-test. The Wilcoxon signed-rank test is appropriate here since the metrics (particularly the efficiency metrics) contain outliers and are not necessarily normally distribute" - This is unclear to me. What is being compared pair-wise? Why? A reference to point the reader to info about a Wilcoxon signed-rank test would be good too.

We agree that the statistical significance analysis we included was not clear in the first draft, and not altogether necessary for the paper. In updating our paper to compare both models with the climatological benchmark, we have removed this (formerly Table 2) in the manuscript. See General Response Results section for more information.

l202. "median values" - Why are only medians discussed here? How meaningful is that on a 5000+ sample?

In our updated analysis, although we still sometimes provide the median for a quantitative point-of-reference, we now include CDFs (Cumulative Density Functions) of the KGE results for (i) the models and the climatological benchmarks, (ii) the Reference versus Non-Reference classification of the gages, and (iii) the 4 regions. See the figures in the General Response Results section for more information.

l206. "indicating that they are tracking similarly in terms of overall performance" - This may need to be a more nuanced. Because these correlations are calculated on ranks and not actual metric scores, I think all this indicates is that these models are similar in where they tend to do relatively better and worse (within their own 5390-member sample). I don't think these ranked correlations indicate that these models are similar in actual performance as measured by the metrics, which is what the text seems to say.

Thank you for this comment. In our revision, we have replaced the Spearman rank correlation with the linear correlation, so to be more consistent with the components of KGE. As such, we have removed this sentence.

l209. "these three popular efficiency metrics are providing very similar information in terms of overall performance assessments" - Again, I think this may need to be a bit more nuanced. What I believe these correlations show is that relative ranks are similar for these three metrics. In the .csv files I can see that there are still quite large differences in the actual scores on the three metrics. I would suggest to rephrase this paragraph.

Thank you for this comment. In our revision, we have removed the NSE and logNSE from the manuscript, so as to focus more on KGE and its comparison with the climatological benchmark. As such, we have removed this sentence.

l216. "Figure 2" - Why is the x-axis in this figure capped at KGE = -0.25? Looking at the data in the .csv files I see that KGE scores go as low as KGE = -306 for the NWM, and KGE = -158 for the NHM. This

suggests that there is a lot of rather poor model performance that's not shown in this figure. Should that not be discussed as well in a paper intended to set a baseline for model performance?

This is a good point, and as indicated in our General Response, we are now comparing with climatological benchmarks, including the mean annual flow KGE benchmark, i.e., −0.41 (Knoben et al. 2019). We have adjusted our x-axis to include this in Figure 2, 3, and 4 (see General Response).

l219. "Table 4 bins the KGE scores" - A similar question can be asked here: why are these bins defined with a lower bin of KGE < 0.2? There seems to be a lot of variety in model performance below this arbitrary threshold. More generally speaking, what can be learned by binning the data in this way that is not obvious from a figure with four CDFs (one CDF each for west, central, southeast and northeast)? These KGE bin boundaries seem quite arbitrary to me and mask any variety within the bin. It might be cleaner to replace this table with CDFs per region instead.

Thank you for this point and suggestion. We have added a new figure that shows the CDFs by region (Figure 4) and removed the previous Table 4 (see General Response).

l231. "Relatively good performance is seen in the Southeast" - This paragraph uses fairly arbitrary thresholds to discuss the KGE performance of both models (e.g., anything with KGE < 0.2 is considered poor performance; KGE > 0.8 is implicitly treated as a boundary above which everything is similarly good). Previous publications argue that efficiency scores such as NSE and KGE cannot be viewed in isolation but need to be compared to some form of baseline model, so that one can judge if these NSE/KGE scores are in fact poor or good for a given location (e.g. Seibert, 2001; Schaefli & Gupta, 2007; Pappenberger et al., 2015; Seibert et al., 2018). NSE includes such a benchmark by design (i.e. the mean annual flow - but this is often criticized as being too easy to beat). KGE does not include such a benchmark and therefore needs some other way to provide context. Work using the CAMELS catchments (Knoben et al., 2020) uses a seasonal cycle benchmark and suggests that for certain locations even KGE > 0.9 could be considered a basic requirement for models rather than being indicative of an exceptionally well-performing model. I think the KGE scores discussed in this paragraph need to be given some context, so that there is some objective reason to qualify a given KGE score as "poor", "good" etc. Presenting these scores in isolation does not help the reader understand what kind of model performance they indicate.

The same comment applies to the following paragraphs as well. The presented numbers need some context that gives the reader an objective reason to decide whether those numbers are indicative of good or bad model performance.

Knoben et al.: doi.org/10.1029/2019WR025975

Pappenberger et al.: doi.org/10.1016/j.jhydrol.2015.01.024

Schaefli & Gupta, 2007: doi.org/10.1002/hyp.6825

Seibert, 2001: doi.org/10.1002/hyp.446

Seibert et al.: doi.org/10.1002/hyp.11476

We appreciate the reviewer's comments here, as well as the literature suggestions. As seen in the general response, we have taken this comment to heart, and are now comparing with the climatological seasonal benchmark following Knoben et al. (2020). We have also added the suggested literature to provide more background for our work.

l244. "It is noticeable that many of the sites are in the tails" - I find this hard to grasp from just looking at this figure. Adding a small histogram to the bottom left corner might help.

In our revision, this sentence has been removed, as we have removed this Figure (formerly Figure 4) and now show the difference between the in Kling–Gupta efficiency (KGE) from the maximum model (i.e., the maximum from the NWMv2.1 or the NHMv1.0) minus the seasonal benchmark based on the average day-of-year flows. See General Response Results for more information.

l315. "here we provide a lower benchmark to gauge the evolution of the NWMv2.1 and NHMv1.0" - This sentence seems to suggest that this publication is mainly intended to benchmark future development of the NWM and NHM. Would a technical report not be a more appropriate venue for this? The kind of information presented in this paper seems useful to those actively working with the NWM or NHM, but may be of somewhat limited interest to the wider hydrological audience.

Thank you for this comment, as we noted in our General response, we appreciate the suggestions to compare with a climatological KGE benchmark to make this of greater interest to the wider community. See General Response for more information.

l317 "The baseline can provide an a priori expectation for what constitutes a "good" model." - I respectfully disagree. This baseline shows the current performance of the NWM and the NHM but it provides no objective reason for calling either a good model. For example, the mean annual flow (NSE = 0; KGE = -0.41) is often used as a rudimentary threshold for model performance. The .csv files with metric values show that the NWM does not outperform the mean annual flow as a predictor in 23% of gauges if NSE is used, and 14% of gauges if KGE is used. Similarly, the NHM does not outperform a mean annual flow in 24% of cases if NSE is used, and 12% of cases if KGE is used. To make the statement that these results are a priori expectations for what constitutes a good model, a much more in-depth comparison of both models against a range of statistical benchmarks (e.g., mean annual flow, seasonal cycle, persistence) and existing model results across this domain (e.g. any number of results based on the CAMELS data, NLDAS [10.1029/2011JD016051], global models [10.5194/hess-24-535-2020]) is needed.

Thank you for this comment. We have revised our paper to compare both models against the climatological KGE benchmarks, including mean annual flow and mean/median daily of year flows). We also appreciate these additional references, and have added them to our Introduction (see General Response).

l336. "Results helped to identify potentially missing processes that could improve model performance. PBIAS results showed that for both models, simulated streamflow volumes are overestimated in the West region, particularly for the sites designated as Non-Reference. One primary reason for this may be that water withdrawal for human use is endemic throughout the West and neither model has a thorough representation of these withdrawals. Furthermore, neither model possesses significant representations for lake and stream channel evaporation which, through the largely semi-arid west, can

constitute a significant amount of water "loss" to the hydrologic system (Friedrich et al., 2018). Lastly, nearly all western rivers are also subject to some form of impoundment. Even neglecting evaporative, seepage and withdrawal losses from these water bodies, the storage and timed releases of water from managed reservoirs can significantly alter flow regimes from daily to seasonal timescales thereby degrading model performance statistics at gaged locations downstream of those reservoirs" - Upon reading this I cannot help but wonder if PBIAS values were needed at all to determine that these models might be improved by accounting for human water use and the presence of lakes & reservoirs. These seem fairly obvious processes to me when one is working with "two models that have been developed to assess water availability and risks in the United States". Should this even be listed as a discussion/conclusion point, instead of being presented as a known a-priori limitation of these models?

In our revision, we have added more in our Discussion around this point. We note that as model development moves towards including human systems, the benchmark results could potentially provide a more concrete goal for "how much" improvement would be needed to adopt a management module. This is of increasing interest as the hydrologic modeling community grapples with how to account for the anthropogenic influence on watersheds, especially since most studies to date focus on minimally disturbed sites. It is also interesting to see that PBIAS is the component that is most useful for this aspect of model diagnostics. Recently, McMillan (2019) links hydrologic signatures to specific processes using only streamflow and precipitation. Interestingly, McMillan (2019) does not find many signatures that relate to human alteration; however, in this paper, streamflow bias metrics are found to be useful in this regard.

l357. "state-of-the-art" - Without intending to disparage the work that undoubtedly has already gone into creating these models, calling them state-of-the-art seems an overstatement if neither of these water resources assessment tools has a way to account for human interaction with the water cycle.

This has been removed.

l354. "Identifying a suite of metrics has an element of subjectivity, but our aim was to identify an initial set of metrics that can be applied to a wide variety of science questions (e.g., see Table 1.1 in Blöschl et al. 365 2013) and that build on standard practices for evaluation of model application performance within the hydrologic community" - As indicated earlier, with 7 out of 9 metrics focusing on bias I find this set of metrics too limited for even an initial set. Of course there is some subjectivity in selecting metrics, but there is also some existing understanding of which statistical properties of hydrographs might be relevant to look at, how those might be captured in streamflow signatures, and how those signatures might be used to explain how well a model simulates certain, specific processes. This current selection of metrics seems too ad-hoc to me and some deeper literature searching would likely result in a set of metrics with a much wider applicability.

Thank you for this suggestion; in addition to our General Response, we provide this draft excerpt to be added to the Discussion: "Identifying a suite of evaluation metrics has an element of subjectivity, but our aim was to focus on streamflow magnitude, since the purpose of the model evaluation effort was for water availability applications. However, magnitude is only one aspect of streamflow, and different metrics for other categories (e.g., frequency, duration, rate of change, etc) could be more appropriate for addressing specific scientific questions or modeling objectives. Recently, McMillan (2019) links hydrologic signatures to specific processes using only streamflow and precipitation. Interestingly, McMillan (2019) does not find many signatures that relate to human alteration; however, in this paper, streamflow bias metrics are found to be useful in this regard."

l576. "Table 1" - It would be helpful if equations were added to each row here. The ratio metrics are currently difficult to interpret for the reader, because they cannot know whether these are calculated as sim/obs or obs/sim without looking into other references.

We have added the equations to the manuscript, and see next response.

l576. "Table 1" - Why are these bias metrics capped at (-)100?

This is the range for PBIAS as we define it; we have added the equation to the paper to make this clear (and equations for PBIAS_HF and PBIAS_LF to the Supplemental):

Percent bias (PBIAS) is calculated as (Zambrano-Bigiarini 2020):

$$PBIAS = \frac{\sum_{t=1}^{N}(S_t - O_t)}{\sum_{t=1}^{N} O_t}$$

where observed flow is $O$, simulated flow is S, and t = 1, 2,… N is the time series flow index.

l642. "Reference (Ref, n= 1,115) and Non-Reference" - A brief explanation of what reference/non-reference means would be helpful. This could be a summary of lines 186-189).

Ref and Non-Ref are now defined in section 3.1. Data, and used consistently throughout.

**Technical corrections**

l162. "modeled and observed" - Is there a word missing that should come after "observed"?

We have added "streamflow" after.

l197. "Using daily observations and simulations from the NWMv2.1 (Towler et al., 2022a) and NHMv1.0 (Towler et al., 2022b) hydrologic modeling applications" - The way the Towler et al. references are inserted in the text implies that they contain the daily time series of observations and simulations, but in reality these references include only the 9 metrics for each gauge. Suggest to clarify this.

This has been clarified.

l204. "the differences are statistically significant given the large sample size" - Why are some values bold in the NWM column and others in the NHM column? Shouldn't they be bold in both or neither?

We have removed the statistical significance analysis from the paper.

l230. "you move" - consider replacing with "one moves"

This has been replaced.

l241. "better and worse" - is there some text missing here that indicates compared against what these models do better or worse?

This sentence has been removed in light of our Major Revision.

l403. "References" - This list is not entirely in alphabetical order.

Thank you, we will check the references for alphabetical order.

l557. "https://10.5066/P9DKA9KQ" - Has this link been inserted correctly? When I click it it attempts to take me to a local file location instead of the link the text suggests this is. Unsure if this problem is on my end only, but the link in the Towler reference above this one seems to work fine for me.

This has been updated and should be https://doi.org/10.5066/P9DKA9KQ

l644. "Figure 2" - these figures are quite small. Stacking the subplots vertically would give more space to each figure.

For the updated 2 panel plots, we now stack the plots vertically.

l673. "Figure 8" - these figures are quite small. Stacking the subplots vertically would give more space to each figure.

For the updated 2 panel plots, we now stack the plots vertically.

l687. "Figure 11" - these figures are quite small. Stacking the subplots vertically would give more space to each figure.

For the updated 2 panel plots, we now stack the plots vertically.

---

## Author Comment (AC4)

**RC2**: 'Comment on hess-2022-276', Robert Chlumsky, 29 Sep 2022

I have completed my review of the paper "Benchmarking High-Resolution, Hydrologic Performance of Long-Term Retrospectives in the United States", Erin Towler et al. The paper presents a benchmark statistical design for the evaluation of process-based hydrologic models over large spatial and temporal scales, and is applied to evaluate the National Water Model v2.1 application of WRF-Hydro and the National Hydrological Model v1.0 of the Precipitation-Runoff Modeling System.

The paper itself is relatively straightforward in methods and application, including a description of both models, description of the metrics selected for evaluation and the presented comparison of the two models using the metrics selected. The paper draws a number of appropriate conclusions regarding the relative performance of the models spatially and based on flow regime, and is overall very well written and logically presented.

Regarding the comments on paper type, the paper aligns largely with a Technical Report format, though the additional discussion and interpretation of results help move it towards a Research Article.

A number of additional comments and concerns are presented here to help improve the paper.

**General Reponse to Referee #2**: We thank the Referee for these comments. We note that we provide a General Response for All Referees, which should be read first. We also respond here to each individual comment in this point-by-point response. As noted in the General Response, we have made major revisions to the (i) Introduction, (ii) Evaluation Approach (iii) Discussion and (iv) Results; this has made our study now of more general interest, and suited to be a HESS Research Article; we appreciate the Referee comments towards this end.

**General Comments**

1. In the Introduction, mention of previous studies that have addressed the 5,390 USGS gages used in this study would be relevant (have any studies used all of these gages as well?)

To our knowledge, this is the first time these 5390 gages have been used in this type of comprehensive evaluation. We point out that we have substantially revised the Introduction, please see General Response.

2. Introduction - It would also be worth mentioning other datasets that have been commonly used in larger-scale benchmark and model intercomparison studies, such as the MOPEX (Duan et al., 2006) and CAMELS dataset (Addor et al., 2017). The Mai et al. (2022) GRIP-GL comparison would also be worth mentioning in the list of recent benchmark and model intercomparison studies.

We appreciate these suggested references and have added them to our revision. Please see General Response for the changes to the Introduction.

3. Introduction - Any previous studies benchmarking these two hydrologic models would be worth mentioning in the last introductory paragraph (lines 73-81), or mention that this is the first study benchmarking these two models specifically.

Although there have been some internal evaluation efforts, they have not been published. We now mention that this is the first time publishing the benchmark results for these two models, in the Discussion: "The presented analysis documented model performance for two large-sample, high-resolution hydrologic models over a long-term period. To our knowledge, this is the first time that these models have been evaluated so comprehensively, as this analysis included 5390 gages, both impacted and non-impacted by human activities."

4. Line 210 – are these three metrics providing very similar information for overall performance assessment in general, or simply because these models are similar and that happens to be the case in this study only? I would be surprised if this conclusion was generalized for very different hydrologic models, and I think this should be carefully rephrased to not overgeneralize from the limited model comparison (i.e. 2 similar models) presented in this study.

As indicated in our General Response, we removed the NSE and logNSE metrics from the suite. As such, we have removed the lines in question from the manuscript.

5. Reference to Knoben et al. (2019) on what a baseline KGE performance is may be useful in interpreting the results, since 0.2 seems somewhat artbitrary. The Knoben et al. paper suggests -0.4 is a more comparable threshold to the NSE=0 interpretation, so perhaps some justification or rationale for using 0.2 is warranted.

We agree with this comment, and as indicated in our General Response, we are now use the KGE > –0.41 as the mean flow benchmark (Knoben et al. 2019), as well as computing the interannual mean/median benchmark values as in Knoben et al. (2020).

6. Table 4 – the bolding pattern is confusing to me, since it is meant to represent the maximum number (percent) of sites by KGE category (?), though the Northeast has two bolded numbers, and in the Central region the minimum number is bolded. Similar bolding patterns continue in other Tables and seem to be at least non-intuitive.

Thanks for this feedback. In our revision, we have removed most of the tables, and have replaced them with CDFs that better show the differences between models, regions, and classification. See the Results in the General Response.

7. Table 6 – I would suggest a summary column with the average metric across regions to help summarize the results, similar to how Table 5 summarizes results for Ref and nonref sites. This would have some duplication with Table 5 but I think it is still worth including here as an additional column

Thank you for pointing this out. Similar to the previous comment, we have removed this table in light of our updated manuscript. Please see updated Results in the General Response.

8.  Figure 4 and lines 241-247 – I thnk that screening the models with poor initial performance from Figure 4, perhaps as a separate figure, would be more meaningful than comparing relative model performance between a KGE of 0.0 and -0.05. In either case, the models likely don't capture enough of the observed behaviour for a modeler to care which is better, and this inhibits interpretation of Figure 4 in identifying any real differences between the models. It seems the models will be similar in any case, but I would filter results first.

Agreed, we have taken this suggestion to heart and adopt this approach of screening the models with poor performance relative to the KGE benchmark. Please see General Response.

9.  Line 268 – I think this statement is actually incorrect, since the lower variability at managed (non-reference) sites should already be normalized by comparison to observed data. My interpretation of this is that the ideal rSD is 1.0, and rSD below 1.0 indicates that the model underestimates the variability of flow. In both cases the models underestimate the variability of flow, in particular for reference sites relative to non-reference or managed sites. This suggests the models do better at capturing general changes in flow rather than sudden ones in unregulated reference sites perhaps. There is more interesting interpretation to add in this section.

We have revised the Results (see General Response), and have revised our examination of the rSD results.

10. Line 277-279 – this can be compared with the GRIP-GL study results (Mai et al., 2022) to discuss general trends in Great-Lakes areas

Thanks for this suggestion. We appreciate pointing out the Mai et al., 2022 paper, and have added it to our Introduction, specifically where we discuss studies that include both impacted and non-impacted gages (See General Response). Further, given that we revised our study to compare the NHM and NWM CONUS-wide models with climatological seasonal benchmark, it would have broadened the scope too much to further compare directly with outputs from other model studies.

11. Line 295 – general comment but an actual histogram plot of the information in Supplemental Table 2 would likely convey this information much better and would aid the discussion. A simple histogram of frequency vs binned PBIAS_LF, and either facet or colour code each of the four regions on one plot would greatly aid the discussion

Agreed, we have removed Supplemental Table 2 and have revised our results to better convey the information, both in terms of new CDF plots and by filtering our results based on the climatological benchmark. We have also added histograms of some of the metrics (PBIAS, PBIAS_LF, PBIAS_HF, and rSD) for both models as separate figures in the Supplemental Material, which are referenced in the updated Results (See General Response). For the benefit of the reviewer, we include the histogram of PBIAS as a preview of what will be included in the finalized Supplemental Material:

[Figure]

**Supplemental Figure 4: Normalized histograms of PBIAS for National Water Model v2.1 (NWMv2.1, top) and National Hydrologic Model v1.0 (NHMv1.0, bottom), for all sites (left) and for sites where the model's KGE score is less than the average day-of-year flow benchmark (right).**

12. Line 341 – it would be worth elaborating on the value of the passive lake/reservoir representation in the model relative the none. It is interesting that the model with the passive representation (NWMv2.1) does seem to perform slightly better than the NHMv1.0, though it is unclear if that is the reason why or what the improvement in performance would be with a better representation of reservoir operations. This would require some segmentation based on catchments with 'significant' reservoir controls, which is not included in this study, though worth discussing briefly here.

Agreed, these differences in performance were only slight, and in our revision, we focus less on the differences between the NHM and NWM. Given the overall manuscript changes, we don't add on to this point in the manuscript, but have added to the draft Discussion: "As model development moves towards including human systems, the benchmark results could potentially provide a more concrete goal for "how much" improvement would be needed to adopt a management module. This is of increasing interest as the hydrologic modeling community grapples with how to account for the anthropogenic influence on watersheds, especially since most studies to date focus on minimally disturbed sites."

13. Line 355 – the NWMv2.1 is described to perform better in high-flow-focused metrics than the NHMv1.0. This discussion should be expanded to how this could likely have

been known from the model setup initially, since running the model on hourly or subdaily timesteps and aggregating will very likely produce better performance for peak flow metrics than a model that is run at a daily timestep, therefore this result should not be a surprise. This is touched on by mentioning that the latter model is designed for water availability, but I think this point should be emphasized.

We have expanded on this, especially in light of new results showing the underestimation of PBIAS in Central by the NHM, we now note in the updated draft Discussion: "Another interesting difference in PBIAS was seen in the Central US, where the NHMv1.0 is underestimating volumes at underperforming sites. As detailed in the model descriptions, the model applications are run at different temporal scales: NHMv1.0 is run daily, whereas NWMv2.1 is run hourly and aggregated to daily. One hypothesis is that some precipitation events that are occurring on sub-daily scales, like convective storms, may be missed, or the associated runoff modes (Buchanan et al. 2018). Similarly, while both models tend to underestimate high flows (PBIAS_HF) and variability (rSD), this is more pronounced for the NHMv1.0, which is in line with this hypothesis. The model applications showed differences in PBIAS_LF, with the NWMv2.1 overestimating low flows, whereas while the NHMv1.0 both over- and under-estimated them it was less extreme. It can be noted that both models used in the applications benchmarked here have only rudimentary representation of groundwater processes. Additional attributes (e.g., baseflow or aridity indices) could be strategically identified to further understand these model errors and differences. Model target applications, which drive model developer selections for process representation, space and time discretization, and calibration objectives, also have a notable imprint on the performance benchmarks. The NWMv2.1, with a focus on flood prediction and fast (hourly) timescales, shows better performance in high-flow-focused metrics, while the NHMv1.0, designed for water availability assessment and slower (daily) timescales, shows better performance in low-flow-focused metrics."

14. Conclusion – the concluding paragraph ends rather abruptly, a short one or two line paragraph at the end to tie off the accomplishments of the paper and goals for future studies would help to transition the conclusion.

Thanks for this suggestion. We have added this as our updated Discussion final paragraph: "In closing, this paper uses the climatological seasonal benchmark as a threshold to screen sites for each model application. While this fit with the purpose of this study, the metrics for NWMv2.1 (Towler et al. 2022a) and NHMv1.0 (Towler et al. 2022b) are available for all sites (Foks et al. 2022); these can be analyzed and/or screened as needed. In the future, it would also be useful to extend the analysis beyond streamflow to other water budget components to assess additional aspects of model performance."

**Technical Comments:**

15. I was under the impression that CONUS was an acronym for contiguous United States (not conterminuous), though I suppose the definitions are practically the same

Thank you, this has been changed to contiguous.

16. Links in lines 92-93 should be properly cited instead of providing raw urls

This citation has been updated.

17. Line 168 – I would rewrite this paragraph slightly to something like: "Three additional hydrologic signatures are included which evaluate performance based on different parts of the flow duration curve (FDC) for high, medium, and low flows. The definitions for these hydrologic signatures as used in this study are consistent with those from Yilmaz et al. (2008). The bias of high flows…" This will help the readability of the section, otherwise the reader is left wondering which metrics you are porting in from Yilmaz until the whole section is read.

Thank you, this has been edited as suggested. We note that we now only include PBIAS_LF and PBIAS_HF (we have removed PBIAS_FDC).

18. Line 201 – "…for all 5,390 cobalt gages …". If these will be called cobalt gages in the paper, this should be used throughout the paper after its definition for consistency

We have removed the term cobalt gages throughout the manuscript.

19. Line 221 – the line "Both models also have many sites with poor performance" – this can be quantified and merged with the next line, as many sites in a large sample study could mean 100 or 1000. Both models in fact have 30% of their sites with a KGE below 0.2, which is a lot of models with very poor performance (KGE below 0.2 is likely an 'unusable' or 'untrustworthy' model for most applications)

These lines have been updated now that our analysis includes a comparison to the climatological KGE benchmark. See General Response.

20. Line 361 – link should be properly cited

In revising our Discussion, this has been removed.

**References**

Thank you for these references, they have been added.

Addor, N., Newman, A. J., Mizukami, N., and Clark, M. P.: The CAMELS data set: catchment attributes and meteorology for large-sample studies, Hydrol. Earth Syst. Sci., 21, 5293–5313, https://doi.org/10.5194/hess-21-5293-2017, 2017.

Duan, Q., Schaake, J., Andréassian, V., Franks, S., Goteti, G., Gupta, H. V., et al. (2006). Model parameter estimation experiment (MOPEX): An overview of science strategy and major results from the second and third workshops. Journal of Hydrology, 320(1–2), 3–17. https://doi.org/10.1016/j.jhydrol.2005.07.031

Knoben, Wouter & Freer, Jim & Woods, Ross. (2019). Technical note: Inherent benchmark or not? Comparing Nash-Sutcliffe and Kling-Gupta efficiency scores. Hydrology and Earth System Sciences Discussions. 1-7. 10.5194/hess-2019-327.

Mai, J., Shen, H., Tolson, B. A., Gaborit, É., Arsenault, R., Craig, J. R., Fortin, V., Fry, L. M., Gauch, M., Klotz, D., Kratzert, F., O'Brien, N., Princz, D. G., Rasiya Koya, S., Roy, T., Seglenieks, F., Shrestha, N. K., Temgoua, A. G. T., Vionnet, V., and Waddell, J. W. (2022):The Great Lakes Runoff Intercomparison Project Phase 4: The Great Lakes (GRIP-GL) Hydrol. Earth Syst. Sci., 26, 3537–3572. Highlight paper. Accepted Jun 10, 2022.

---

## Author Comment (AC5)

RC3: ['Comment on hess-2022-276'](), Anonymous Referee #3, 03 Oct 2022

This is a review of the manuscript "Benchmarking High-Resolution, Hydrologic Performance of Long-Term Retrospectives in the United States" by Towler et al. The manuscript compares the performance of two large-scale hydrologic models in estimating streamflow by comparing against observed streamflow at gauges across continental United States (CONUS). The performance is evaluated using a number a metrics that are commonly used in streamflow evaluation. The manuscript is well-written and easy to follow. The effort to create benchmarks for CONUS scale streamflow prediction models is commendable, necessary, and of interest to this journal and the hydrologic community. However the metrics presented here are commonplace and the evaluation/benchmarking workflow is not novel. My biggest criticisms of the study are regarding the consistency of comparing two model outputs (major comment 2) and the use of calibration gauges in evaluation (major comment 3).

The manuscript can still be considered for publication provided the authors sufficiently address my concerns. I, therefore, recommend Major Revision.

**General Reponse to Referee #3**: We thank the Referee for these comments. We note that we provide a General Response for All Referees, which should be read first. We also respond here to each individual comment in this point-by-point response. As noted in the General Response, we have made major revisions to the (i) Introduction, (ii) Evaluation Approach (iii) Discussion and (iv) Results.

**Major comments:**

1. Introduction: The Introduction is missing a comprehensive review of current literature and needs improvement to further clarify the hurdles being overcome by this study and bring out its novelty. Specifically, the last paragraph should have a few sentences summarizing how it is building on previous studies and what shortcomings are being overcome in this specific study. Additionally, for studies mentioned in L 48-65, please mention their drawbacks and how this study aims to overcome them. Also, review of studies regarding statistical design of large-sample benchmarks and intercomparisons has been ignored. The authors should also clarify how the benchmark statistical design used in this study compares to previous studies where large sample intercomparison and/or benchmarking have been carried out. Finally, the National Hydrologic Model is mentioned for the first time in the manuscript in L 75 when the authors are specifying the objectives of the study. The authors should introduce the two models briefly in the Introduction while also mentioning the reasons behind choosing these two specific models.

Thank you for these comments. We have revised the Introduction, please see the General Response. In short, we note that a drawback of most studies to date is that they are evaluating smaller, minimally-impacted basins (and we have added additional studies here, including Duan et al. 2006 using MOPEX, and Knoben et al. (2020) using CAMELS). Nevertheless, most river

basins are impacted by human activities. These impacted basins also need to be benchmarked; especially as model development moves to include human systems. While there are some studies that have begun to address this globally (Arheimer et al. 2020); in Great Britain (Lane et al. 2019); Great Lakes Region (Mai et al. 2022); and for 1 year over the Central US (Tijerina et al. (2021); this has not been done for a long-term retrospective over the entire CONUS. This comprehensive evaluation of a long-term retrospective over the CONUS, using both impacted in addition to non-impacted sites, is our first contribution. We now note in the updated draft Discussion that to our knowledge, this is the first time that these models have been evaluated so comprehensively, Further, our second, related, contribution is facilitated by adopting and extending another suggestion to our paper, which was to provide performance context for our models. We now compare our two models to a climatological benchmark of KGE based on the interannual mean for each site, as in Knoben et al. (2020), and extend this by using it as a threshold to further scrutinize the metric results. Please see the General Response for details.

2. L 113: NWM produces hourly streamflow using hourly atmospheric forcings whereas NHM produces daily streamflow using daily forcings. The hydrologic processes in the watersheds are simulated at different temporal scales (hourly vs daily) by the two models. Additionally, the many USGS gauges record 15-minute streamflow data. NWM can produce hourly streamflow and takes into account changes in hydrologic variables throughout the day. Averaging out higher resolution (hourly) streamflow timeseries produced using higher resolution (hourly) forcing to a coarser resolution is not the equivalent of simulating streamflow at a coarser resolution (daily) from coarse resolution (daily) forcings due to the non-linear nature of hydrologic processes. As such, is the comparison of the streamflow produced at two different temporal scales a consistent and fair comparison?

Thanks you for this comment. We note that in our original preprint, we did focus on the comparison between the NHM and NWM, whereas in the updated manuscript, we now compare both models with a climatological benchmark. However, we agree that we need to be transparent about the model differences, including the different temporal scales. This comes out in a new point brought up in the draft updated Discussion, where we now note: Another interesting difference in PBIAS was seen in the Central US, where the NHMv1.0 is underestimating volumes at underperforming sites. As detailed in the model descriptions, the model applications are run at different temporal scales: NHMv1.0 is run daily, whereas NWMv2.1 is run hourly and aggregated to daily. One hypothesis is that some precipitation events that are occurring on sub-daily scales, like convective storms, may be missed, or the associated runoff modes (Buchanan et al. 2018; https://doi.org/10.1002/hyp.13296). Similarly, while both models tend to underestimate high flows (PBIAS_HF) and variability (rSD), this is more pronounced for the NHMv1.0, which is in line with this hypothesis. The model applications showed differences in PBIAS_LF, with the NWMv2.1 overestimating low flows, whereas while the NHMv1.0 both over- and under-estimated them it was less extreme. It can be noted that both models used in the applications benchmarked here have only rudimentary representation of groundwater

processes. Additional attributes (e.g., baseflow or aridity indices) could be strategically identified to further understand these model errors and differences. Model target applications, which drive model developer selections for process representation, space and time discretization, and calibration objectives, also have a notable imprint on the performance benchmarks. The NWMv2.1, with a focus on flood prediction and fast (hourly) timescales, shows better performance in high-flow-focused metrics, while the NHMv1.0, designed for water availability assessment and slower (daily) timescales, shows better performance in low-flow-focused metrics.

Buchanan, B., Auerbach, D.A., Knighton, J., Evensen, D., Fuka, D.R., Easton, Z. Wieczorek, M., Archibald, J.A., McWilliams, B., and Walter, T.: Estimating dominant runoff modes across the conterminous United States, Hydrological Processes, 32: 3881–3890, https://doi.org/10.1002/hyp.13296, 2018.

3. Calibration: What was the calibration period for the two models? It is unclear from the text if gauges used in calibration were also part of evaluation. If the calibration period overlapped the evaluation period (October 1, 1983, to December 31, 2016), then the gauges used for calibrating either of the models should be removed from the set of gauges used for benchmarking the models. Including these gauges will introduce biases in the evaluation process.

The calibration periods differed for the two models. For the NWMv2.1, the calibration period was from water years 2008 – 2013, and 2014-2016 was used for validation. For the NHMv1.0, the calibration period included the odd water years from 1981-2010, and the even water years from 1982-2010 were used for validation. This has been added to the model descriptions. As such, the calibration period did overlap with the evaluation period for the calibration gages, but it was not consistent between the models. We acknowledge the reviewer's point, but note that our approach fit with our objectives, which was to evaluate the long-term performance of both models at the same sites and time periods. The same technique was adopted in the MOPEX study (Duan et al. 2006). Further, there has been recent research activity in calibration. In particular recent studies suggest updating calibration techniques to use the full available data period and to skip model validation entirely (Shen et al. 2022; https://agupubs.onlinelibrary.wiley.com/doi/10.1029/2021WR031523 ). We have added sentences to this effect in the updated Discussion section.

Shen, H., Tolson, B. A., & Mai, J. (2022). Time to update the split-sample approach in hydrological model calibration. Water Resources Research, 58, e2021WR031523.

4. The study also includes gauges near the coast in the evaluation scheme. USGS gauges do not measure streamflow directly, rather the water surface elevation (WSE) is measured and the WSE is converted to streamflow using rating curves. Gauges near the coast can experience backwater from coastal surge traveling up the river and/or tides. In such cases, the rating curve for converting WSE to streamflow are violated and streamflow

readings are highly erroneous. As such, should gauges near the coast be included in the evaluation scheme? Additionally, both NWM and NHM do not take into account the interaction between the river and sea/ocean.

We do not include tidally influenced gages in this analysis, though this was an interesting point we had not considered. For the benefit of the reviewer, we note that we did speak to the USGS team, and they indicated that backwater effects are accounted for in USGS gauging procedures (tidal or otherwise). We believe that most are not stage-discharge gages but rather index-velocity gages which allow negative velocities. They indicated that the tougher issue is how to handle negative velocities (flows) in a hydrologic (water only downhill) type model; which would be challenging to either the NHM or NWM. In some previous work, when all the GAGESII references gages were analyzed, negative flows were very rare at the daily time step; recalling there were only two gages in Florida that had such. In short, we don't believe this to be a major issue in this analysis.

5. L 327-330: The authors should discuss why these areas are exhibiting poorer/better performance for both the models. They have a done good job of explaining the behavior of PBIAS in L 335-348 and need to similarly delve deeper into the potential causes of the behavior in the efficiency metrics for these regions.

Thank you for these comments. We have updated our Results significantly to delve deeper into the performance of both models as compared to the climatological benchmark; please refer to the General Response, Results section.

6. The authors need to discuss the limitations of this study and future work at the end of the manuscript in more detail. The limitations of the study extend beyond the subjectivity in choosing the performance metrics and their sensitivities. This could be a separated section or can be a continuation of the Results and Discussions.

We have updated the Discussion, including points raised in major comments #2 and #3. Further, we have added this as our updated Discussion final paragraph: "In closing, this paper uses the climatological seasonal benchmark as a threshold to screen sites for each model application. While this fit with the purpose of this study, the metrics for NWMv2.1 (Towler et al. 2022a) and NHMv1.0 (Towler et al. 2022b) are available for all sites (Foks et al. 2022); these can be analyzed and/or screened as needed. In the future, it would also be useful to extend the analysis beyond streamflow to other water budget components to assess additional aspects of model performance."

**Minor Comments:**

7. Title: is it really the United States if Alaska and the US territories have not been included? Should it be CONUS instead?

We have amended the title to be contiguous United States

    8.  L 177: The study uses 5,390 gauges and 5,389 of those are in GAGES II. So, there is just one gauge that was not part of GAGES II?

Yes, only one was not part of GAGES II, but it fit all the other criteria so it was included.

    9.  L 191: "For statistical significance …" – statistical significance of what?

We agree that in the first draft of the paper this was confusing and unnecessary to the evaluation analysis. In updating our paper to compare both models with the climatological benchmark, we have removed this in the manuscript, see General Response.

    10. L 350: refer to the appropriate table/figure

This has been fixed.

    11. Table 3 can be moved to supplementary information. KGE and NSE (and logNSE) are expected to behave somewhat similarly given their formulations. So this table does not convey anything particularly novel or important.

We have removed Table 3, and further now focus the paper on KGE (we have removed NSE and logNSE); see General Response.

    12. Figure 2: There can be further subplots showing the CDF of KGE for the two models by region. This will be more informative than Table 4 which can then be moved to supplementary information.

Thank you for this comment – we agree and have changed several of the Tables to be CDFs, which we agree are much clearer; see the Results in the General Response.

    13. Figure 4: Just a suggestion, with there being so many points, it is hard to discern a trend or behavior from the figure. It might help to have region-wise or HUC-unit-wise medians color coded across CONUS. See Figure 8 in https://doi.org/10.1016/j.jhydrol.2022.127470 as an example.

Thank you for this suggestion. We note that we have removed Figure 4, and are now filtering the results by sites that underperforming relative to the climatological KGE benchmark (which reduces the number of points).

    14. Please adjust the font size in the figures to make sure the legends, subplot number and lat/long are easily readable (Figures 3, 8, 11)

Some of the figures have changed, but we have adjusted the font sizes in the updated figures to make them easily readable. Further, we hope this will be helped by now stacking the figures vertically (rather than side-by-side).

---

## Author Response (AR2)

**Author Response to Editor and Reviewer**

**Editor Comments:**

Dear authors,

After subsequent feedback from the reviewers, I'd like to recommend moderate revisions to your manuscript. I strongly encourage you to consider the major recommendation from the reviewer for incorporation into your manuscript.

Overall, the reviewers agree that the manuscript has improved. The reviewers also indicate that there are still changes that can be made to elevate the impact of this study.

I look forward to receiving and reviewing your updated manuscript.

**Author Response to Editor:**

Dear Editor,

We appreciate the opportunity to revise our manuscript. We have addressed Referee #1's major recommendation, which is to apply the R package from the Clark et al. (2021) paper to compute the KGE uncertainty for each of the gages. To demonstrate the uncertainty in the KGE estimates, we have generated a new figure which we include in the Supplemental Material. Further, the generated KGE uncertainty data for all of the gages was added to each of the respective data releases: for NWMv2.1 they have been added to Towler et al. 2023a and for NHMv1.0 they have been added to Towler et al. 2023b. Please see details in our response to Referee #1, below.

**Anonymous Referee #1 Report:**

*Suggestions for revision or reasons for rejection (will be published if the paper is accepted for final publication)*

I would like to thank the authors for their substantial overhaul of this manuscript in response to reviewer suggestions. I think these changes have made the paper more relevant, particularly due to the benchmarking approach the authors applied. I think there is value in publishing this paper as a means to encourage more detailed benchmarking of model performance and to encourage a shift in community thinking away from "look at how high my efficiency scores are" and towards deliberate assessment of model weakness (and hence areas of possible model improvement).

If I have one major comment to make it is that the authors note that "one limitation of this study is that it does not consider the sensitivity of the KGE to sampling uncertainty, which can be large for heavy-tailed streamflow errors (Clark et al., 2021)". This would be a straightforward limitation to address, because this Clark et al. paper also points to an R package that can be used to compute the KGE uncertainty in an easy manner. I believe this would lift the paper to a higher level.

Thank you for the feedback, and we have addressed your major comment. Following from Clark et al. (2021), we have run the gumboot package (Clark and Shook 2021) for all the gages to

compute the KGE uncertainty. Using the KGE uncertainty outputs, we have generated a new figure which we include in the Supplemental Material (see later in this response for more details on this figure). Further, the KGE uncertainty data was added to the existing data releases: for NWMv2.1 they have been added to Towler et al. 2023a and for NHMv1.0 they have been added to Towler et al. 2023b. These have been added as individual csv files (with their own metadata); they include the outputs from the gumboot *bootjack()* function for the KGE, including the standard error of jacknife, standard error of bootstrap, the 5th, 50th and 95th percentiles of the estimates, the jackknife score, the bias of jackknife, the bias of bootstap, the standard error of jackknife after bootstrap (Clark and Shook 2021). Each new csv file includes all 5390 gages from Foks et al. (2022), but includes NAs where there is not sufficient data to compute the *bootjack()* function. For the NHMv1, uncertainty estimates could be computed for 5312 out of 5390 gages, and for the NWMv2.1, 5288 of the 5390 gages. Following from Clark et al. (2021), the uncertainty estimates of the KGE estimates were plotted for the CAMELS dataset (Addor et al. 2017) from each model using the gumboot *ggplot_estimate_uncertainties()* function. The figure has been added as Figure 11 of the Supplemental Material. The figure shows that the bootstrap and jackknife yield similar estimates, and that there is uncertainty in KGE for both models, similar to what is found in Clark et al. (2021). We have amended the Discussion as follows:

"Clark et al. (2021) point out that it is important to characterize the sensitivity of KGE to sampling uncertainty, which can be large for heavy-tailed streamflow errors. Using bootstrap methods (Clark et al. 2021), uncertainty in the KGE estimates for this study were computed (Towler et al. 2023a, 2023b) and are illustrated in Supplemental Figure 11."

The below figure and caption were added to the Supplemental Material:

[Figure]

**Supplemental Figure 11. Estimates of uncertainty in the KGE estimates for the CAMELS (Catchment Attributes and Meteorology for Large-sample Studies) basins (Addor et al. 2017) using the gumboot package (Clark and Shook, 2021) in R (R Core Team, 2021) for the National Water Model v2.1 (NWMv2.1; top) and National Hydrologic Model v1.0 (NHMv1.0; bottom). Quantification of the uncertainty is obtained from 2x standard error estimates obtained using jackknife and bootstrap estimates, as well as intervals computed as the difference between the 95[th] and 5[th] percentiles of the bootstrap samples (see Clark et al. 2001 for details). The figure shows the uncertainty in the KGE estimates, with the bootstrap and jackknife showing similar estimates for both models. KGE uncertainty estimates for the full set of gages in this study (Foks et al. 2022) are included in Towler et al. (2023a, 2023b).**

**References:**

Addor, N., Newman, A. J., Mizukami, N., and Clark, M. P.: The CAMELS data set: catchment attributes and meteorology for large-sample studies, Hydrol. Earth Syst. Sci., 21, 5293–5313, https://doi.org/10.5194/hess-21-5293-2017, 2017.

Clark M. and Shook, K: Package 'gumboot: Bootstrap Analyses of Sampling Uncertainty in Goodness-of-Fit Statistics', available online at: https://cran.r-project.org/web/packages/gumboot/index.html (last access March 6, 2023), 2021.

Clark, M. P., Vogel, R. M., Lamontagne, J. R., Mizukami, N., Knoben, W. J. M., Tang, G., et al.: The abuse of popular performance metrics in hydrologic modeling, Water Resour. Res., 57, e2020WR029001. https://doi.org/10.1029/2020WR029001, 2021.

Foks, S.S., Towler, E., Hodson, T.O., Bock, A.R., Dickinson, J.E., Dugger, A.L., Dunne, K.A., Essaid, H.I., Miles, K.A., Over, T.M., Penn, C.A., Russell, A.M., Saxe, S.W., and Simeone, C.E.: Streamflow benchmark locations for conterminous United States, version 1.0 (cobalt gages): U.S. Geological Survey data release, https://doi.org/10.5066/P972P42Z, 2022.

R Core Team: R: A language and environment for statistical computing. R Foundation for Statistical Computing, Vienna, Austria, available at: https://www.r-project.org (last access: 4 May 2022), 2021.

Towler, E., Foks, S.S., Staub, L.E., Dickinson, J.E., Dugger, A.L., Essaid, H.I., Gochis, D., Hodson, T.O., Viger, R.J., and Zhang, Y.: Daily streamflow performance benchmark defined by the standard statistical suite (v1.0) for the National Water Model Retrospective (v2.1) at benchmark streamflow locations for the conterminous United States (ver 3.0, March 2023): U.S. Geological Survey data release, https://doi.org/10.5066/P9QT1KV7, 2023a.

Towler, E., Foks, S.S., Staub, L.E., Dickinson, J.E., Dugger, A.L., Essaid, H.I., Gochis, D., Hodson, T.O., Viger, R.J., and Zhang, Y.: Daily streamflow performance benchmark defined by the standard statistical suite (v1.0) for the National Hydrologic Model application of the Precipitation-Runoff Modeling System (v1 byObs Muskingum) at benchmark streamflow locations for the conterminous United States (ver 3.0, March 2023): U.S. Geological Survey data release, https://doi.org/10.5066/P9DKA9KQ, 2023b.

Beyond that I only have a handful of minor editorial suggestions:

Line 157. "(Falcone 2011)" - There's a comma missing here.
This has been added.
Line 197. "select KGE scores" - Out of curiosity, what prompted the choice of -0.06 (and 0.50 and 0.75) as thresholds? The reasoning for this might be added to the text.
The reason for the -0.06 is already present in the text, but we now specify that one should look at Figure 2 (the KGE CDF plot) to see that -0.06 is where the KGE CDF curves intersect. We have added (change in bold): "**From Figure 2, it can be seen that t**The CDFs for the models intersect with the AvgDOY curve at a KGE score of about -0.06; at this value, 19%-20% of the sites perform worse in terms of KGE using the model simulation, whereas above this value the model simulations perform better than AvgDOY." The other values (0.50 and 0.75) were selected to provide select quantitative values to go with the figure.
Line 208. "it" - consider replacing with "a gauge".
This has been added.
Line 227. "Southeast" - It may be good to specify which states area meant here. Looking at Florida, performance seems uniformly quite poor.
We have made the following changes (additions in bold): "Relatively good performance is seen in **most of** the Southeast, **but performance tends to be poor or mixed in Florida**."
Line 247. "you look" - consider replacing with "one looks".
This has been changed.
Line 295-297. "Lane et al. ... in the models." - This sentence might be better moved to Line 289, before "One likely reason ...", with "One likely reason ..." replaced by "This is a likely

explanation in our case as well, because water withdrawal ... ". I think this would improve the flow of this section.

We have incorporated these suggestions.

Table 5. I would suggest to use 1 decimal place for all percentages listed in column "n (%)" so that the sums all cleanly add to 100%.

This has been done.

Figure 3. It's interesting that the simple data-based benchmark model has virtually identical performance in Reference and Non-reference gages, whereas the models clearly struggle a lot more with the Non-reference gages than the Reference gages. This may indicate that, even though many gages are managed, their flows are (generally speaking) about as seasonally stable, and hence about as predictable by this simple model, as the flows at unmanaged gages. This may point to an opportunity to use really simple post-processing tools to simulate water management in the Non-ref basins. No real comment beyond this or any need for the authors to do something with this, I just found it interesting to mention.

Agreed it is an interesting finding.

Figure 5, 6. Is there a particular reason to set the 5 color bins in these plots (and possibly others) at [-Inf, 0.2], [0.2, 0.4], etc.? Having a lower bin of [-Inf, -0.41] might make more sense, as this gives some reason to lump everything below those numbers together. The remaining 4 bins might be evenly divided as [-0.41, -0.06], [-0.05, 0.30], etc.

We have changed the color bins for Figure 5 and 6 as suggested; see below:

[Figure]

**Figure 5: Kling–Gupta efficiency (KGE) based on daily streamflow at U.S. Geological Survey (USGS) gages for (A) National Water Model v2.1 (NWMv2.1) and (B) National Hydrologic Model v1.0 (NHMv1.0). The Map Source: (Grannemann, 2010; Natural Earth Data, 2009; ESRI, 2022a; ESRI, 2022b).**

[Figure]

**Figure 6: Kling–Gupta efficiency (KGE) based on daily streamflow at U.S. Geological Survey (USGS) gages using seasonal benchmark from average day-of-year flows (AvgDOY). Map Source: (Grannemann, 2010; Natural Earth Data, 2009; ESRI, 2022a; ESRI, 2022b).**